# Phospho-signaling couples polar asymmetry and proteolysis within a membraneless microdomain in *Caulobacter crescentus*

Yasin M. Ahmed [1], Logan M. Brown[2], Krisztina Varga [2] & Grant R. Bowman[1]✉

Asymmetric cell division in bacteria is achieved through cell polarization, where regulatory proteins are directed to specific cell poles. In *Caulobacter crescentus*, both poles contain a membraneless microdomain, established by the polar assembly hub PopZ, through most of the cell cycle, yet many PopZ clients are unipolar and transiently localized. We find that PopZ's interaction with the response regulator CpdR is controlled by phosphorylation, via the histidine kinase CckA. Phosphorylated CpdR does not interact with PopZ and is not localized to cell poles. At poles where CckA acts as a phosphatase, dephosphorylated CpdR binds directly with PopZ and subsequently recruits ClpX, substrates, and other members of a protease complex to the cell pole. We also find that co-recruitment of protease components and substrates to polar microdomains enhances their coordinated activity. This study connects phospho-signaling with polar assembly and the activity of a protease that triggers cell cycle progression and cell differentiation.

Many bacteria divide asymmetrically, generating two distinct cell types that can have vast differences in gene expression, morphology, and other behaviors[1–3]. This basic form of multicellularity was very likely to have arisen more than a billion of years before the emergence of eukaryotic life[4]. In rod-shaped bacteria, asymmetry is generally achieved through cell polarization, where distinct sets of regulatory proteins are directed to opposite cell poles, and subsequent cell division generates daughter cells that inherit different sets of regulatory factors and therefore different cell fates[5,6]. These systems depend on mechanisms that distinguish one pole from the other, and in many cases the associated molecular processes that drive this organization are not clear.

In Alphaproteobacteria, one of the key polarization factors is the polar organizing protein PopZ, which is required for the polar localization of many cell fate regulators[7]. In vitro, PopZ self-assembles into oligomeric forms that undergo higher-order assembly into fibrils and larger structures[8,9]. Under some conditions, these assemblages behave as liquid-liquid phase-separated protein condensates and have the ability to interact with client proteins[10,11]. PopZ exhibits similar assembly and recruitment activities when expressed in *Escherichia coli*,

where it accumulates as a single cytoplasmic focus[12–14]. In its natural context, PopZ and its associated proteins form three-dimensional structures that abut the membrane at cell poles[15]. *Caulobacter crescentus* PopZ is localized to both poles through most of the cell cycle, yet several of the regulatory proteins that depend on it for polar localization exhibit transient, unipolar localization[16–18].

The N-terminal domain of PopZ is an interaction hub that binds to at least eleven different client proteins and recruits them from the cytoplasm to polar microdomains[12,19]. Including indirect interactions, PopZ serves as localization determinant for large interconnected protein networks[12,16–18]. For example, the protease ClpP and its partner ClpX are indirectly recruited to polar microdomains through interaction with the regulatory factor CpdR, which binds directly to PopZ[12,17]. CpdR is a cell cycle-regulated adaptor protein that delivers substrate proteins to ClpXP for timely degradation[20]. Some ClpXP substrates require one or more additional adaptor proteins, namely RcdA and PopA, and these are recruited to cell poles through interaction with PopZ at the same stage of the cell cycle as CpdR, which is known as the swarmer to stalked cell transition[21]. Thus, ClpP, ClpX, up to three adaptors, and the substrate proteins themselves[22,23] are all co-recruited

[1]Department of Molecular Biology, University of Wyoming, Laramie, WY, USA. [2]Department of Molecular, Cellular, and Biomedical Sciences, University of New Hampshire, Durham, NH, USA. ✉e-mail: grant.bowman@uwyo.edu

by PopZ into polar microdomains, where they become concentrated relative to bulk cytoplasm.

The question of how co-localization and concentration in PopZ microdomains affects the recruited proteins' activities has not been conclusively resolved. Owing to the limited size of PopZ structures and the relative abundance of available clients, many PopZ-associated proteins are only partially localized in polar microdomains[24,25]. This raises the question of whether the levels of concentration and co-localization at cell poles are large enough to influence cell physiology. One study found that three different pole-localized ClpXP substrates were degraded more quickly in a ΔpopZ strain than in wildtype[26], a phenotype that could occur if polar microdomains have an inhibitory effect on the activity of protease complexes. However, this is not easily reconciled with the observation that both ClpXP-mediated proteolysis and the polar accumulation of ClpXP complexes occur at the same stage of the cell cycle.

Two other unresolved questions concern the regulation of transient localization to cell poles and the mechanisms by which one PopZ microdomain is differentiated from another. These questions are exemplified by the members of the ClpXP proteolysis complex, which are localized to only one of the two poles, and only during the swarmer to stalked cell transition. Notably, both the CpdR and RcdA adaptor proteins are direct binding partners of PopZ[12,19], which suggests that their interaction with the polar hub is subject to some form of regulation.

CpdR is a two-component response regulator whose phosphorylation state is regulated by the histidine kinase CckA, via an intermediary phospho-transfer protein ChpT. CckA exhibits kinase activity through the majority of the cell cycle, when CpdR and ClpXP protease complexes are neither pole-localized nor active in degrading substrates[17,20,27]. CckA works as a phosphatase during the swarmer to stalked cell transition, when CpdR is localized to one pole and ClpXP is active. CckA binds directly to PopZ and is localized to both cell poles through most of the cell cycle, where its activity is controlled through the influence of asymmetrically localized upstream regulators[28].

Taking these observations together, we hypothesize that CckA kinase/phosphatase activity is a switch that regulates PopZ's interaction with CpdR and the consequent recruitment of other members of ClpXP proteolysis complexes. In this work, we tested that hypothesis by modulating CpdR phosphorylation levels in C. crescentus, in a reconstituted E. coli system, and in vitro. We also asked if the concentration of ClpXP complex members at the cell pole could influence the overall rate of proteolysis for the entire cell, using a combination of cell imaging and computational modeling. The results explain how CpdR phosphorylation and de-phosphorylation influences polar asymmetry and cell differentiation during the C. crescentus cell cycle.

## Results

### CpdR-YFP localization is correlated with its phosphorylation state during the cell cycle

In wildtype C. crescentus, polar localization of CpdR-YFP temporally coincides with the time that it is dephosphorylated by CckA-ChpT phospho-transfer during the swarmer-to-stalked transition[17,27] (Fig. 1a). We used cell cycle synchronization to quantify the relationship between CpdR-YFP localization and phosphorylation. After isolating cells at the swarmer cell (G0) stage, we counted the fraction of cells exhibiting polar CpdR-YFP localization (Fig. 1b) and quantified the relative levels of phosphorylated versus unphosphorylated CpdR-YFP over the course of the cell cycle (Fig. 1c). The highest and lowest degrees of polar localization temporally corresponded with the highest and lowest fractions of unphosphorylated CpdR-YFP, respectively. After cell division, CpdR-YFP remained diffuse in swarmer progeny but exhibited polar localization in stalked progeny, and correspondingly, we observed an intermediate level of CpdR-YFP phosphorylation in this heterogenous population (Fig. 1c). We conclude that there is a strong positive correlation between CpdR-YFP dephosphorylation and polar localization during the C. crescentus cell cycle.

### CckA kinase influences CpdR-YFP phosphorylation and its co-localization with PopZ

We asked if we could influence CpdR-YFP phosphorylation and localization by expressing mutant variants of CckA[29] with differing levels of kinase/phosphatase activity from a plasmid (Fig. 1d–h). When we expressed a hyperactive-kinase (H+) form of CckA (G319E), the ratio of phosphorylated to unphosphorylated CpdR-YFP was relatively high, and the protein did not localize to cell poles. When we expressed a kinase deficient (K−) form of CckA (H322A), the ratio was substantially lower, and cells exhibited robust polar foci. We also observed transient localization of CpdR-YFP at the division plane, which likely arises from CpdR's interaction with ClpXP protease complexes at this location[30]. To determine if polar localization is indirectly controlled by CckA signaling, we expressed a mutant form of CpdR-YFP (CpdR$_{D51A}$-YFP) that cannot be phosphorylated[17]. Whether or not the hyperactive-kinase form of CckA was expressed in this context, CpdR$_{D51A}$-YFP exhibited polar localization, and moreover, was usually localized to both of the cell poles in cells that also had a bi-polar distribution of mChy-PopZ (Fig. 1g, h). These results show that the kinase/phosphatase activity of CckA is closely correlated with the phosphorylation state of CpdR and its co-localization with PopZ at C. crescentus cell poles.

### CpdR phosphorylation affects interaction with PopZ in E. coli

We reconstituted CpdR phosphorylation and studied polar localization in E. coli to determine whether CpdR's phosphorylation state affects its interaction with the polar organizing protein PopZ. To do this, we co-expressed CpdR-GFP and mChy-PopZ together with the phospho-transfer protein ChpT and then added either a wildtype, kinase deficient, or hyperactive-kinase variant of CckA (Fig. 2a). Experiments were performed in a ΔclpXP mutant background[31] to eliminate the possibility of interactions between C. crescentus CpdR and endogenous protease[20]. Combining ChpT and CpdR-GFP without any CckA resulted in a higher level of CpdR-GFP phosphorylation than CpdR-GFP alone (24% compared to 4%), indicating a background level of host-derived phosphorylation, which has been observed for some other heterologous response regulators in E. coli[32]. We could modulate CpdR-GFP phosphorylation by co-expressing hyperactive-kinase, wildtype, and kinase-deficient variants of CckA, which drove CpdR-GFP phosphorylation levels of 99%, 79%, and 17%, respectively (Fig. 2b). Under these conditions, we assessed interactions between CpdR-GFP and mChy-PopZ by observing these proteins' sub-cellular localization patterns. Cells that expressed an N-terminal truncation mutant of PopZ or lacked mChy-PopZ did not accumulate CpdR-GFP in polar foci (Fig. 2c and Supplementary Fig. 2c), consistent with earlier studies which demonstrate that the N-terminal hub domain of PopZ is responsible for this interaction[19]. In cells expressing full-length mChy-PopZ, the presence or absence of ChpT itself did not affect CpdR-GFP's co-localization with mChy-PopZ in polar foci (Supplementary Fig. 2b). With the further addition of wildtype CckA, we observed a similar degree of CpdR-GFP mChy-PopZ co-localization. In contrast, adding the hyperactive-kinase variant of CckA reduced the frequency of CpdR-GFP polar localization to zero (Fig. 2c). Conversely, strains expressing kinase-deficient CckA exhibited increased co-localization, and we found similarly elevated levels of co-localization when we expressed non-phosphorylatable CpdR$_{D51A}$-GFP with hyperactive-kinase CckA. From these experiments, we conclude that phosphorylating CpdR inhibits its ability to co-localize with polar PopZ foci in E. coli.

### CpdR-PopZ interaction is highly dynamic

We consistently observed that a large fraction of CpdR is diffuse in the cell body, even when it is almost entirely dephosphorylated

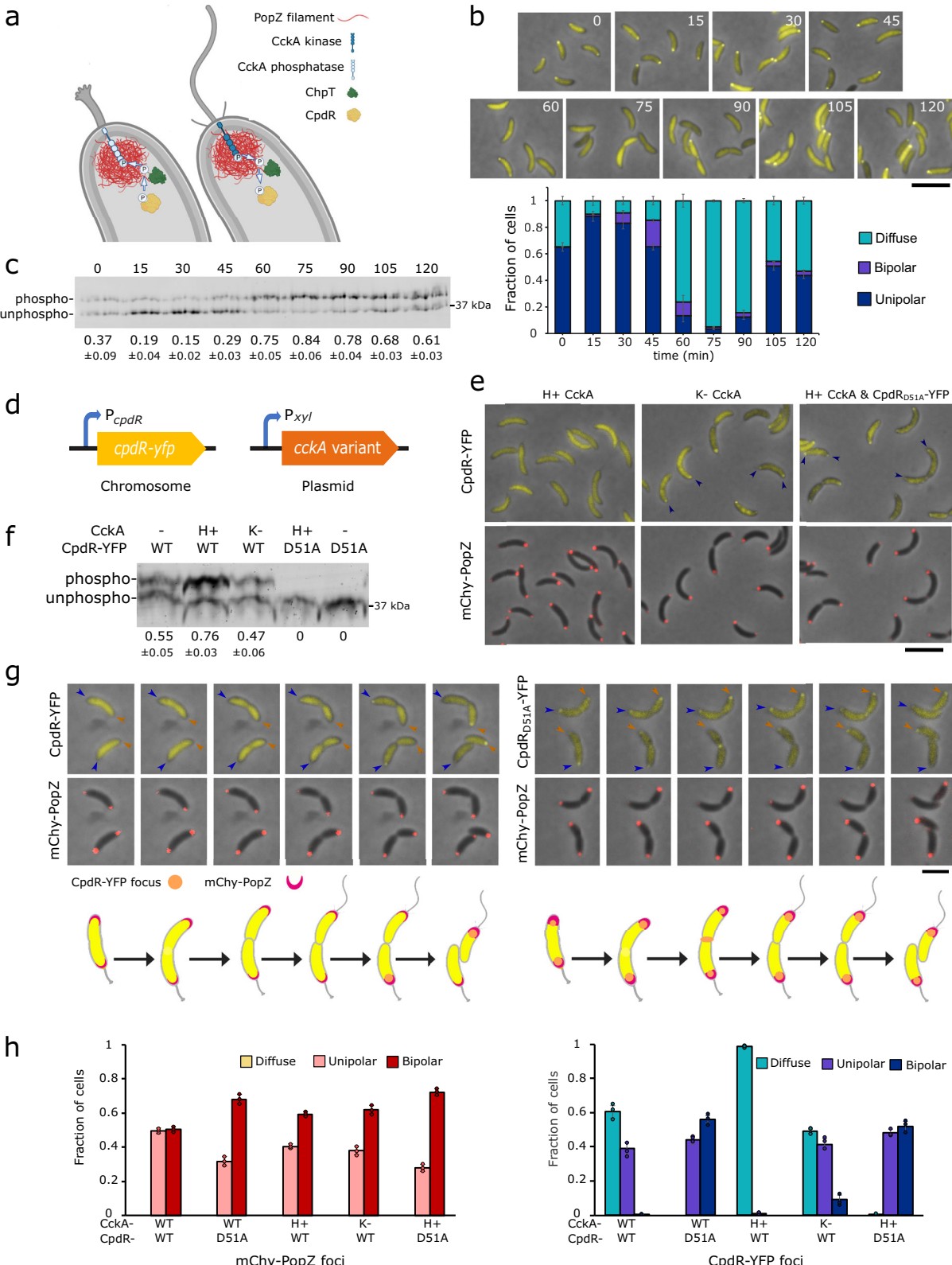

(Figs. 1b,g and 2c). We hypothesized that this is reflective of a weak and therefore highly transient interaction with PopZ, and tested this in FRAP experiments on *E. coli* cells co-expressing CpdR-GFP and mChy-PopZ (Fig. 2d). We found that PopZ-associated CpdR-GFP was rapidly replenished from the cytoplasmic pool after photobleaching, with a half-time of 0.61 s. This was slightly slower than the rate of recovery of CpdR-GFP diffusing through normal cytoplasm, at 0.36 s. During

recovery, a wavefront of CpdR-GFP advanced through PopZ foci (Supplementary Video 1), which could occur if CpdR-GFP molecules are transiently held by interactions with PopZ. To obtain information on the off-rate, we created *E. coli* cells with mChy-PopZ foci at both cell poles. Bleaching CpdR-GFP at one pole resulted in rapid fluorescence recovery, with concomitant rapid fluorescence loss at the opposite pole. By contrast, the recovery rate of mChy-PopZ had a half-life of

**Fig. 1 | Correlations between CpdR's phosphorylation state, polar localization, and co-localization with PopZ. a** CpdR phospho-signaling at stalked (left) and swarmer (right) cell poles, where CckA acts as a phosphatase or kinase, respectively. Graphics: BioRender. **b** *C. crescentus* cells expressing CpdR-YFP were synchronized and, at indicated time points over a cell cycle time course, aliquots were removed for observation. The plot shows the average frequencies of cells exhibiting zero, one, or two polar foci, and the error bar shows the range between the two biological replicates ($n > 300$/replicate/time point, scale bar = 5 μm). **c** CpdR-YFP phosphorylation levels in lysates from (**b**), observed by Phos-tag gel electrophoresis. The average intensities of the phosphorylated bands as a fraction of the sum of band intensities from both biological replicates are provided, along with the range between replicates. The replicate gel is provided in Supplementary Fig. 1a. **d** Genetic modifications for controlling CpdR-YFP phosphorylation. Single-copy *cpdR-yfp* is expressed from the native promoter, multicopy *cckA* variants are expressed from $P_{xyl}$ without xylose induction. **e** Localization of CpdR-YFP or $CpdR_{D51A}$-YFP and mChy-PopZ in different CckA signaling contexts. H+ and K−

signify hyperactive kinase and kinase-deficient forms of CckA expressed from a multicopy plasmid, in addition to CckA expressed from the unmodified *cckA* locus. Arrowheads mark polar localization; scale bar = 5 μm. **f** CpdR-YFP phosphorylation levels in lysates from (**e**), observed using Phos-tag gel electrophoresis. The dash mark indicates no extrachromosomal copies of *cckA*. The average intensities of the phosphorylated bands as a fraction of the sum of band intensities from three biological replicates are provided, along with the standard deviation between replicates. Replicate gels are provided in Supplementary Fig. 1b. **g** Cells expressing CpdR-YFP or $CpdR_{D51A}$-YFP and mChy-PopZ were observed during cell division, using time-lapse microscopy at 15 min intervals. The fluorescence levels of individual panels were adjusted differently to aid visualization. Scale bar = 2 μm; Graphics: BioRender. **h** Average frequencies of cells with diffuse, monopolar, and bipolar fluorescent foci, from strains imaged in (**e**, **g**) ($n > 100$/replicate, bar = standard deviation of three biological replicates). Source data and *n* values for (**b**, **h**) are provided as a Source Data file.

over 3 min (Supplementary Fig. 2d), indicating that associations among the scaffold molecules are more stable.

## Direct interaction between CpdR and PopZ is controlled by phosphorylation

We used solution NMR spectroscopy to determine whether CpdR interacts directly with PopZ. CpdR-GFP was chosen as a ligand because the fusion protein was far more soluble than untagged CpdR at high concentration. The other ligand was C-terminally truncated $PopZ^{\Delta134-177}$, which retains the N-terminal protein–protein interaction domain (Fig. 2c), but lacks the ability to self-assemble[9]. [15]N-enriched $PopZ^{\Delta134-177}$ (50 μM) was mixed with increasing concentrations of unlabeled CpdR-GFP (0–750 μM), and the spectra were analyzed for concentration-dependent changes (Fig. 3a). Most of the $PopZ^{\Delta134-177}$ NMR peaks exhibited no significant changes, even at the highest concentration of CpdR-GFP, indicating residues whose local biochemical environments are not affected by the presence of this protein. A small number of $PopZ^{\Delta134-177}$ residues displayed large chemical shift perturbations and significant peak broadening with increasing concentrations of CpdR-GFP. To ask if these effects are caused by interactions with GFP, we mixed 750 μM GFP with 50 μM of [15]N-enriched $PopZ^{\Delta134-177}$. No significant chemical shift perturbations were observed in this control experiment (Supplementary Fig. 3b), indicating that the large perturbations observed upon the addition of CpdR-GFP were induced by interactions with CpdR. The peaks exhibiting the most significant chemical shift perturbation and broadening are likely indicative of residues that interact directly with CpdR, whereas peaks with moderate effects chemical shifts may correspond to amino acids participating in secondary or indirect binding interactions (Supplementary Fig. 3a). $PopZ^{\Delta134-177}$ residues E7-E23 exhibited the greatest changes, and these results are in good agreement with previous studies that identified the N-terminus of $PopZ^{\Delta134-177}$ as the interaction site for RcdA and ChpT[12,19].

We asked if CpdR can also interact with macromolecular scaffolds comprised of full-length PopZ, and if this interaction can be influenced by CpdR phosphorylation. To do this, we generated phase-separated condensates of purified full-length PopZ[10] and demonstrated that CpdR-GFP but not GFP alone partitions into the condensates (Fig. 3b). Next, we phosphorylated CpdR-GFP by pre-incubating it with acetyl phosphate (AcP), using $CpdR_{D51A}$-GFP as a non-phosphorylatable receiver to control for the presence of AcP (Fig. 3c). We found that CpdR-GFP phosphorylation inhibited the partitioning of CpdR-GFP into PopZ condensates, while the same treatment had no such inhibitory effect on the $CpdR_{D51A}$-GFP control, nor did it affect protein mobility in Phos-tag gel electrophoresis (Fig. 3c, d). These results suggest that CpdR phosphorylation reduced the partitioning of CpdR-GFP into PopZ condensates.

## CpdR phosphorylation influences the localization of RcdA and ClpX

Since ClpX physically interacts with CpdR[17], we hypothesized that we could control the localization of ClpXP complexes (Fig. 4a) by modifying the phosphorylation state of CpdR. To test this, we created strains in which either the ClpX-associated adaptor RcdA[18] or ClpX itself were tagged with GFP and expressed from their endogenous promoters, and in which the native copy of *popZ* had been exchanged with a functional mChy-tagged version[7] (Fig. 4b). In an otherwise wildtype genetic background, RcdA-GFP and ClpX-GFP both exhibited the expected pattern of transient co-localization with polar mChy-PopZ during the swarmer-to-stalked cell transition[18,30] (Fig. 4c, d). In a Δ*cpdR* background, RcdA-GFP was almost completely diffuse, although many cells retained faint polar foci that were difficult to discern and quantify (Supplementary Fig. 4a). In agreement with an earlier report[17], we also observed that CpdR is required for the polar localization of ClpX.

To assess protein localization in cells where CpdR is always dephosphorylated, we replaced endogenous *cpdR* with the $cpdR_{D51A}$ variant. RcdA-GFP was localized to both poles in cells that also had bipolar mChy-PopZ foci, and, in agreement with an earlier report[17], ClpX-GFP showed a similar, though less intensely localized pattern (Fig. 4c, d). To assess protein localization in cells where CpdR is always phosphorylated, we expressed the hyperactive variant of CckA. In these cells, RcdA-GFP and ClpX-GFP were diffuse. When we expressed hyperactive CckA in the $cpdR_{D51A}$ background, RcdA-GFP and ClpX-GFP exhibited bi-polar localization, indicating that CpdR dephosphorylation is a controlling factor in these proteins' polar localization. The differences in these strains' localization patterns suggest that in normal cells, CpdR's phosphorylation state is under strict temporal and spatial control, directing ClpXP complexes to one pole at the swarmer-to-stalked cell transition. Since ClpX is reported to play a key role in RcdA-GFP polar localization[18], CpdR may promote RcdA localization indirectly, by recruiting ClpX to the pole. Additional RcdA may be localized to poles via direct contact with PopZ[19].

## Substrate proteolysis coincides with polar localization

To assess the localization of proteolysis substrates, we induced the expression of YFP-tagged versions of PdeA, TacA, and CtrA RD + 15[33] in wildtype and other genetic backgrounds and observed their localization patterns during the cell cycle. These three substrates were expressed from chromosomally integrated xylose-inducible promoters, and were chosen as representatives of CpdR-dependent, CpdR/RcdA-dependent, and CpdR/RcdA/PopA-dependent classes, respectively. Their localization patterns are consistent with earlier reports on PdeA and CtrA localization[7,34], and add to the literature by showing that YFP-TacA localization depends on RcdA (Supplementary

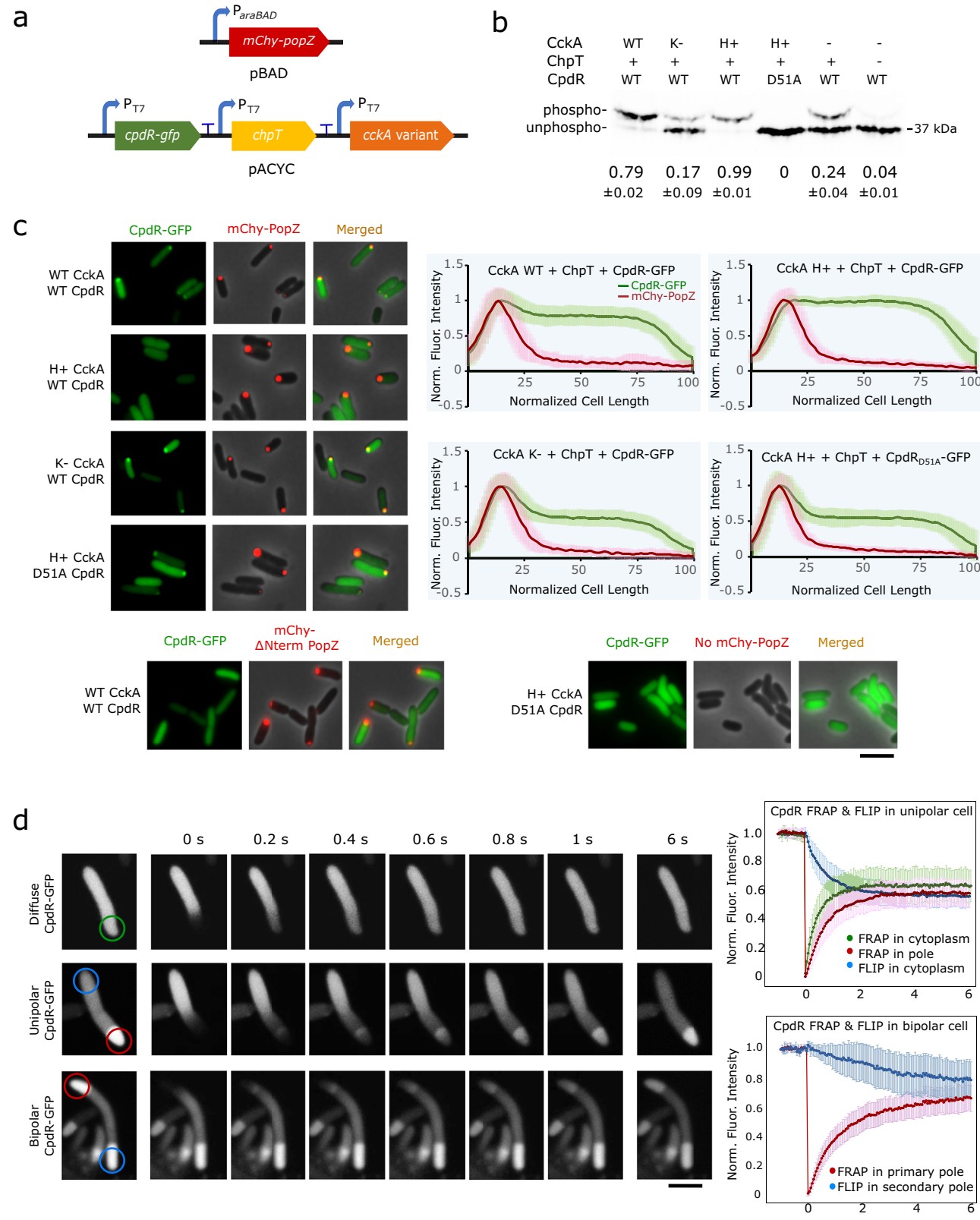

Fig. 4b), and that all three classes of substrates exhibit transient, unipolar localization in wildtype cells and diffuse localization in both Δ*popZ* and Δ*cpdR* backgrounds (Fig. 5a).

We used time-lapse fluorescence microscopy to observe the spatial-temporal relationships between localization and proteolysis for the three substrate proteins (Fig. 5b). In stalked cell progeny, which are programmed to advance immediately to S-phase and subsequent stages of the cell cycle[35–37], all three proteolysis substrates were cleared in the minutes before physical cell separation, and the highest frequency of polar localization was observed during this time period. Since cell separation lags the separation of progeny cells' cytoplasm via inner membrane fusion by several minutes[38], it is likely that

**Fig. 2 | CpdR phosphorylation state influences CpdR-PopZ interactions in *E. coli*. a** Genes for reconstituting CpdR phosphorylation and PopZ interaction in *E. coli*. **b** CpdR-GFP phosphorylation levels in *E. coli* lysates, observed using Phos-tag gel electrophoresis. The average intensities of the phosphorylated bands as a fraction of the sum of band intensities from three biological replicates are provided, along with the standard deviation between replicates. Replicate gels are provided in Supplementary Fig. 2a. For CckA variants, WT = wildtype;

H+ = hyperactive-kinase; K− = kinase-deficient. **c** mChy-PopZ and CpdR-GFP localization in *E. coli* cells. Normalized fluorescence intensities were plotted against cell length (*n* = 60, with 20 cells from 3 biological replicates). Lines trace mean value, shaded regions = SD). Scale bar = 5 μm. **d** FRAP and FLIP assay for CpdR-GFP in *E. coli* cells expressing PopZ. Recovery and loss of fluorescence were plotted against time in seconds (*n* = 20, Lines trace mean value, bar = standard deviation). Scale bar = 2 μm. Source data are provided as a Source Data file.

proteolysis and polar localization occurred soon after compartmentalization. In swarmer cell progeny, which are developmentally delayed relative to stalked cell siblings, YFP-PdeA was often cleared concomitantly or within four minutes of its clearance from stalked cells, which coincided with the highest frequency of polar localization. The majority of YFP-TacA and YFP-CtrA RD + 15 were cleared ~2 min later, when their peak frequency in polar localization occurred. In a Δ*popZ* genetic background, none of the substrate proteins were localized to polar foci (Fig. 5b), and our measurements of YFP-CtrA RD + 15 fluorescence intensity (Supplementary Fig. 5a) suggest that it is cleared at a substantially slower rate than in a wildtype genetic background. Together, these observations provide additional evidence, now for multiple substrates and at high temporal and spatial resolution, that the polar localization of proteolysis substrates is closely correlated with their rapid proteolysis[39].

## PopZ facilitates rapid substrate degradation

To better understand the functional relationship between substrate protein degradation and polar localization, we performed two different types of experiments to assess the degradation rates in wildtype versus Δ*popZ* cells. In the first type of experiment, we blocked new protein synthesis by treating cells with chloramphenicol. Over subsequent time points, a subpopulation of cells in wildtype cultures retained large quantities of substrates for more than 60 min, but this was not observed in Δ*popZ* cultures (Supplementary Fig. 5b, c). We propose that chloramphenicol treatment interferes with the measurement of degradation rates by preventing cell cycle progression in swarmer cells, locking them into a stage of high substrate stability. Δ*popZ* strains, whose cell division is often uncoupled from the cell cycle[7], do not appear to produce significant numbers of this cell type.

In the second type of experiment, we used inducer wash-out to block new substrate protein synthesis, allowing general protein synthesis and the cell cycle to continue (Fig. 5c and Supplementary Fig. 5a). Under these conditions, the three ClpXP substrates that we tested were degraded more rapidly in wildtype cells compared to Δ*popZ*. The difference was higher for substrates whose degradation is mediated by more adapter proteins: CtrA requires three adapters (CpdR, RcdA, and PopA), TacA requires two (CpdR and RcdA), and PdeA requires only CpdR[23]. Taking this together with the observation that the substrates, adaptors, and protease are concentrated in PopZ microdomains during proteolysis (Figs. 1g, 4c, and 5b), we propose that one of the functions of PopZ microdomains is to enhance the assembly of ClpXP-adaptor-substrate complexes.

## A conceptual model for enhanced assembly of proteolysis complexes in PopZ microdomains

PopZ microdomains occupy ~0.5% of the cytoplasm[15], and only a fraction of proteolysis substrates and protease components are found in this relatively small compartment. A question is whether PopZ microdomain could enhance proteolysis complex assembly and activity on a scale that is sufficient to affect the overall rate of degradation for the entire cell. Using the biochemical simulation program Smoldyn[40], we developed a computational model to determine whether the physiological characteristics of the system, in terms of compartment sizes, protein localization, diffusion rates, rate of proteolysis,

and number of substrate molecules, are compatible with the idea that PopZ microdomains enhance proteolysis.

We created a simplified model that considers interactions between two reactants, A and B, which diffuse within the cell and are eliminated after colliding. Using live-cell single-molecule tracking of pole-localized *C. crescentus* proteins to inform particle diffusion rates[38], molecules A and B were allowed to exhibit Brownian motion in PopZ microdomains, but with a slower diffusion coefficient compared to normal cytoplasm. This had the effect of increasing local protein concentration, mimicking the effect of weak interactions with PopZ. In simulated cells with physiologically relevant values for microdomain size and particle diffusion rates, 50% of the reactants were degraded within 1.5 min, compared to more than 5 min in cells that lacked a polar microdomain (Fig. 6a and Supplementary Video 4). Increasing the number of substrate molecules in polar compartments, by either increasing compartment volume or lowering the rate of diffusion at the cell pole, had the effect of increasing reaction rate (Fig. 6a, b). Gains in reaction efficiency were subject to rational limits on microdomain volume and particle behavior (Supplementary Fig. 8a, b). For example, at extremely low diffusion coefficients, particles became "stuck" in polar microdomains and, although highly concentrated, collided less often. Further, concentrating only one of the two reactants in polar microdomains had no beneficial effect on the reaction rate (Fig. 6b).

We also created a more complex model, with physiological concentrations of protease and adaptors (Fig. 6c and Supplementary Video 4). Basing our model on known interactions with PopZ[12], particles representing CpdR and RcdA interacted directly with polar microdomains, and, following in vivo observations[25], they were concentrated in this compartment by reducing their polar diffusion coefficients to 1/40th the cytoplasmic rate. ClpXP, which was modeled as a hexamer[41] that could bind up to six molecules of CpdR, was recruited to polar microdomains indirectly (Supplementary Fig. 8c). We added two model substrates, Sub1 analogous to PdeA in that it interacted with CpdR-primed ClpXP complexes for degradation[20], and Sub2 analogous to TacA in that it required both CpdR and RcdA as adaptors for ClpXP degradation[23]. All protein–protein interactions were given the same association and dissociation rates, which corresponded to low-affinity binding ($K_D = 100$ μM).

With these parameters, 14.4% of the total pool of CpdR was concentrated in polar microdomains (Supplementary Fig. 8c), which approximates our in vivo observations (17.1 ± 4.1% of CpdR-YFP was concentrated in polar foci at the 15 min time point in Fig. 1a). This model mimicked the pattern of substrate degradation in *C. crescentus* swarmer cells in that the single-adaptor substrate was degraded earlier than the dual-adaptor substrate (Fig. 5b). This is a sensible outcome, since three-component assemblies (Sub1, CpdR, and ClpXP) are more likely to form than four-component assemblies (Sub2, RcdA, CpdR, and ClpXP), and the limiting population of ClpXP particles is predominantly occupied by the more frequently occurring substrate complex. We observed that the proteolysis rate of the single-adaptor substrate was not strongly influenced by PopZ, while the time to 75% proteolysis for the dual-adaptor substrate absence was more than doubled in the absence of PopZ. We also found that the sequential timing of substrate degradation was sharpened and the influence of PopZ on Sub1 more closely resembled our in vivo data on PdeA

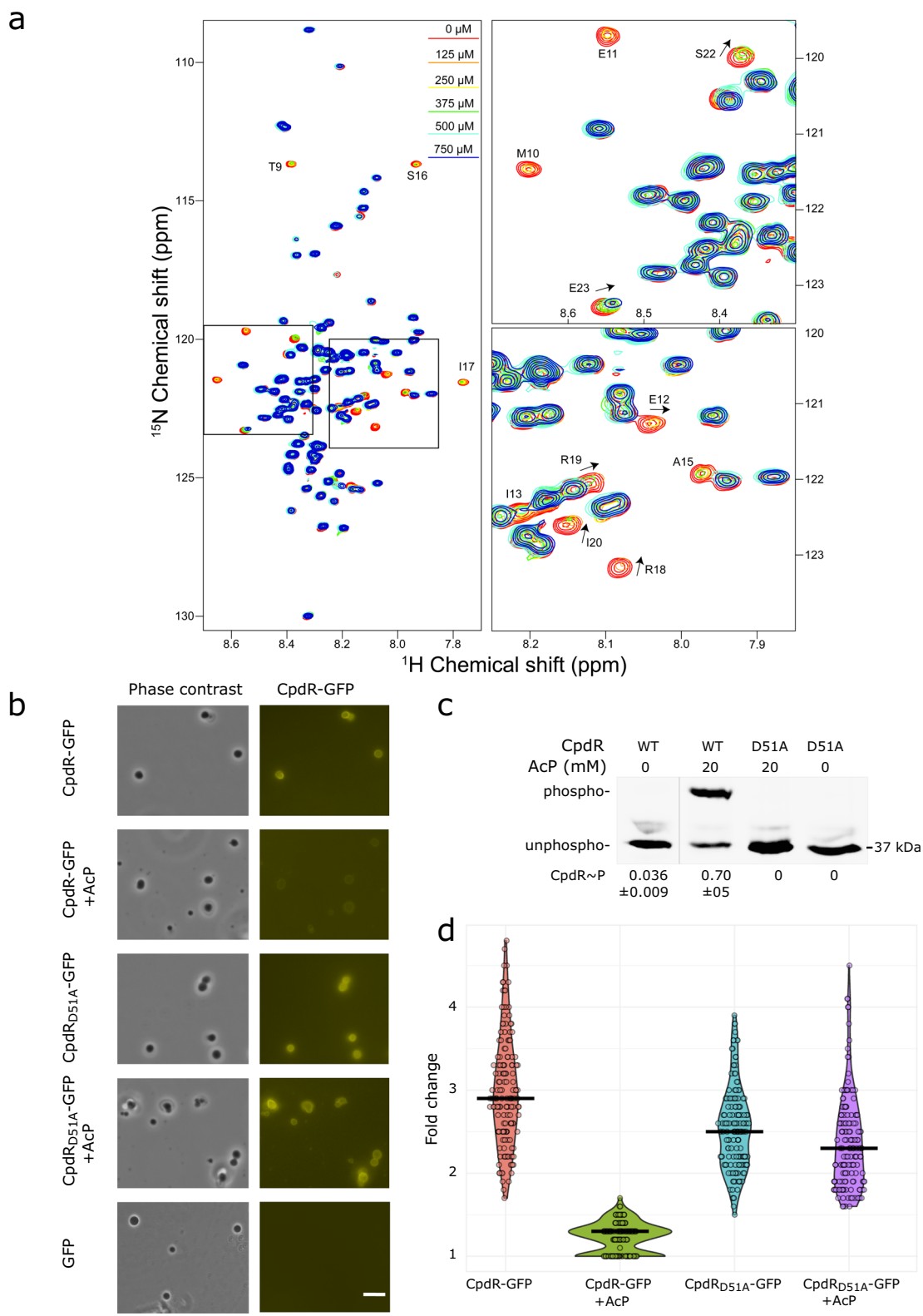

degradation (Fig. 5c) if the model included physical interaction between Sub1 and PopZ, although the evidence for physical interaction between PdeA and PopZ is inconclusive[12]. In conclusion, the simulations provide support for the idea that the assembly and activity of protease complexes can be enhanced when multiple components are concentrated within PopZ microdomains, and that this can occur on a size and time scale that is relevant to proteolysis in vivo.

## Discussion

In this work, we show that CpdR's interaction with pole-localized PopZ is closely correlated with its phosphorylation state (Figs. 1–3), that changing CpdR phosphorylation level through CckA signaling or the expression of the CpdR$_{D51A}$ variant is sufficient to alter the localization of CpdR and associated ClpXP complexes between diffuse and bi-polar (Fig. 4), and that the polar accumulation

**Fig. 3 | Direct interaction between CpdR and PopZ is inhibited by phosphorylation. a** Left: $^1$H-$^{15}$N TROSY-HSQC NMR spectra overlay of 50 μM $^{15}$N-enriched PopZΔ$^{134-177}$ with varying concentrations of CpdR-GFP: 0 μM (red), 125 μM (orange), 250 μM (yellow), 375 μM (green), 500 μM (cyan), and 750 μM (blue). Right: Enlarged regions highlight the changes observed with increasing CpdR-GFP concentration. Residues with the most significant perturbations are labeled, with arrows indicating the direction of the peak shift. **b** CpdR-GFP, CpdR$_{D51A}$-GFP, or GFP alone was incubated with PopZ condensates. Phase contrast and YFP fluorescence channels are shown. +AcP = pre-incubation with acetyl phosphate. Scale bar = 10 μm. **c** CpdR-GFP phosphorylation levels in samples from (**b**), observed using Phos-

tag gel electrophoresis. Figure panel shows a single gel from which irrelevant lanes are removed. The average intensities of the phosphorylated bands as a fraction of the sum of band intensities from 3 biological replicates are provided, along with the standard deviation between replicates. Replicate gels are provided in Supplementary Fig. 3a. **d** Ratios of the CpdR-GFP and CpdR$_{D51A}$-GFP fluorescence intensities within condensates to outside condensates, imaged in YFP channel. Violin plot widths are proportional to the number of data points, bar shows population average. n = 150 condensates per sample (50 per biological replicate). Source data are provided as a Source Data file.

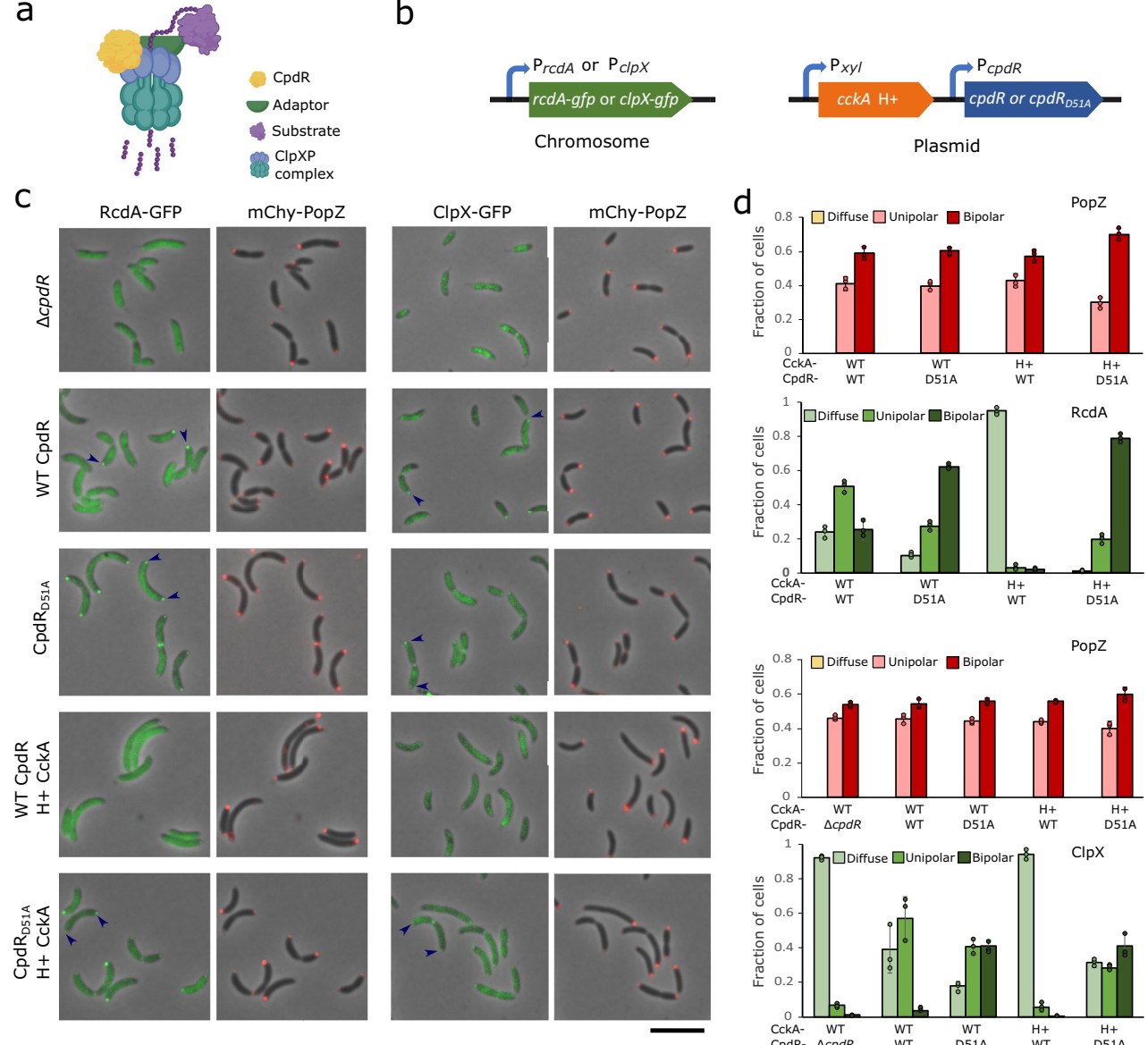

**Fig. 4 | Correlations between CpdR phosphorylation state and the localization of protease components. a** Diagram of CpdR-mediated substrate degradation by ClpXP. Depending on the substrate, an additional adaptor (green) could be RcdA and/or PopA. Graphics: BioRender. **b** Genetic modifications for observing RcdA-GFP or ClpXP-GFP in different CpdR phosphorylation contexts, built in a Δ*cpdR*; *popZ::mChy-popZ C. crescentus* strain background. **c** Localization of RcdA-GFP or ClpX-GFP and mChy-PopZ in different CpdR phosphorylation contexts. Arrowheads mark polar localization; scale bar = 5 μm. **d** The average frequencies of cells with diffuse, monopolar, and bipolar fluorescent foci in (**c**) (n > 100/replicate, bar = standard deviation of 3 biological replicates). Source data and n values for (**d**) is provided as a Source Data file.

of ClpXP substrates is positively correlated with proteolysis (Fig. 5). Together, these results suggest that CpdR-mediated recruitment of ClpXP to polar PopZ subdomains stimulates proteolytic activity (Fig. 6).

This study shows that phosphorylation directly regulates a client protein's interaction with the PopZ hub. An earlier study showed that a chromosome segregation protein, ParA, loses affinity to PopZ after hydrolyzing ATP[42]. Other studies have found that a ClpX adaptor,

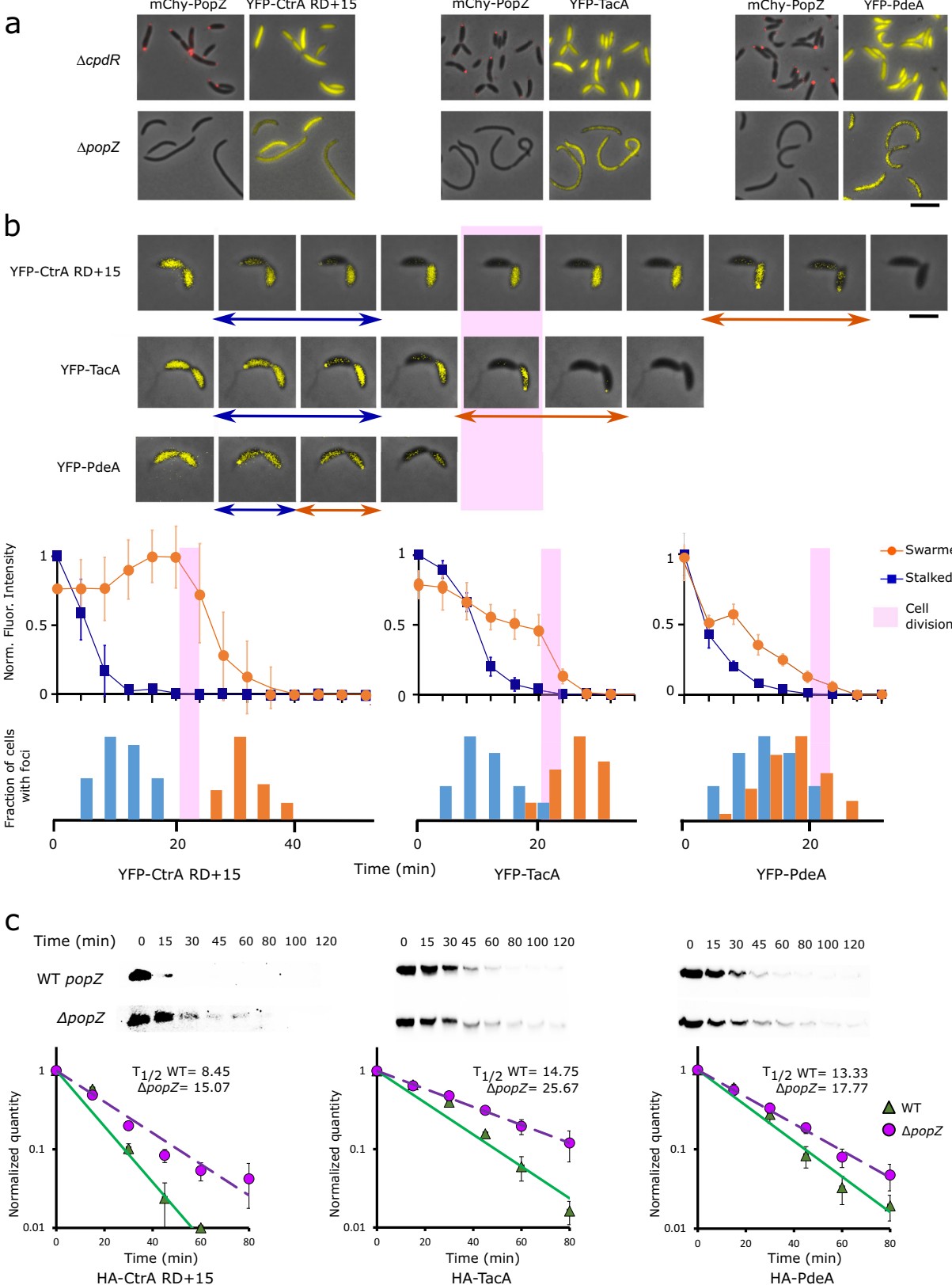

PopA, interacts with PopZ when it is bound to cyclic di-GMP[43,44]. CpdR has a more basal position than PopA in the adaptor hierarchy that determines ClpXP substrate specificity, meaning that it targets a substantially larger number of substrates, including those that are targeted by PopA[23]. A third adaptor, RcdA, interacts directly with PopZ in *E. coli* co-expression experiments but requires dephosphorylated CpdR

(Fig. 4c) or c-di-GMP bound PopA[45] for polar localization in *C. crescentus*, where protein expression levels are lower. We hypothesize that RcdA itself has a relatively low affinity to PopZ that is increased by associating with other PopZ-interacting proteins, forming multi-protein complexes with high avidity. All three ClpX adaptors' interactions with PopZ are post-translationally regulated, suggesting that

**Fig. 5 | Polar localization of CpdR substrates is correlated with increased rate of degradation. a** Localization of YFP-tagged substrates in Δ*cpdR*; *popZ::mChy-popZ* and Δ*popZ C. crescentus* strain backgrounds. Scale bar = 5 μm. **b** Time-lapse images of YFP-tagged substrate localization in a WT *C. crescentus* background, at 4 min intervals. Blue arrows mark frames with foci in stalked cell, orange arrows mark frames with foci in swarmer cell. Pink bar idicates the time of cell separation. After accounting for photobleaching and temporally aligngning the cells with respect to the time of cell separation, average fluorescence intensities for stalked and swarmer cell bodies, normalized to maximum fluorescence intensity, were plotted against time (line graphs, with error bars showing standard deviation, *n* = 20 cells). The fractions of cells ehxhibiting a polar focus, normalized to the highest value observed, were plotted on the same time axis (bar charts). Scale bar = 2 μm. **c** Degradation of HA-tagged proteolysis substrates following inducer wash-out, observed by western blotting with α-HA antibody. Average band intensities from three separate experiments were plotted against time (graphs, bar = standard deviation). Source data are provided as a Source Data file.

this is a key leverage point in the regulation of proteolysis and cell cycle progression.

The mechanisms that control whether CckA works as a kinase or phosphatase are driving forces behind the localization of CpdR and associated members of ClpXP protease complexes. In stalked cells, where PopZ exhibits bipolar localization and CpdR/ClpXP are exclusively at the stalked pole (Figs. 1g, 4c), the differential activity of CckA at opposite poles[29] may be sufficient to support this aspect of polar asymmetry. However, CckA control may not be sufficient to explain all aspects of protease localization and activity. Whereas three different substrates were localized and degraded simultaneously in stalked cell progeny, PdeA was localized and degraded several minutes earlier than TacA and CtrA in swarmer cell progeny (Fig. 5b). Some timing differences could be related to the expression of additional regulatory factors during pole remodeling at the swarmer to stalk transition. CtrA proteolysis requires a polar diguanylate cyclase to activate PopA[34], and TacA proteolysis may require the phosphatase-stimulating activity of SpmY, which is also recruited to the transitioning pole[46].

The term "polar localization" is used here and in most other reports to communicate the idea that a protein is visibly concentrated at the cell poles relative to the bulk cytoplasm. But this term is highly imprecise and potentially misleading, as a protein may exhibit "polar localization" when a small fraction of the total is at a pole. For example, since both cell poles occupy ~0.1% of the total cytoplasm volume, only 1% of a protein that is 10-fold concentrated at cell poles would be pole-localized. Indeed, fluorescence images of many different proteins in *C. crescentus*, including CpdR, substrates, and other members of the ClpXP complex, suggest low levels of actual polar localization. How could a system benefit from having such a localization pattern?

An answer to this question may lie in the fact that many pole-localized proteins receive regulatory cues at a cell pole and carry out their activity in bulk cytoplasm. CpdR is dephosphorylated by CckA at the stalked cell pole, yet its targets are transcriptional regulators, chemotaxis regulators, and other proteins that work outside of PopZ microdomains. Thus, dynamic exchange between pole and cytoplasm is required for translating polar signaling into cytoplasmic activity. This is consistent with the CpdR-PopZ interaction dynamics we observed in *E. coli* (Fig. 2d) and the behavior of other polar regulators in *C. crescentus*[25,47].

Low levels of polar localization coupled with highly dynamic exchange can have a substantial influence on entire populations of molecules on the whole-cell scale (Fig. 6a, b). Applying this to adaptors, substrates, and ClpXP complexes, we propose that their concentration in PopZ microdomains increases the frequency of intermolecular collisions and therefore enhances multiprotein complex assembly and overall proteolysis rate (Fig. 6c, d). Faster ClpXP-dependent proteolysis could make the cell cycle more robust by sharpening phase changes at the swarmer-to-stalked transition.

## Methods
### Bacterial plasmid and strain construction
All plasmids were constructed by Gibson cloning method, and descriptions are provided in Supplementary Table 1. Bacterial strains are listed in Supplementary Table 2. Fluorescent proteins mCherry (mChy), msfGFP or eYFP are described in FPbase.org.

***E. coli* vectors.** Among the two multiple cloning sites of pACYC-duet (Invitrogen); from site 1 we used NcoI and NotI restriction sites for cloning *cpdR-gfp* and from site 2 we used NdeI and XhoI restriction sites for cloning *chpT*. For both CpdR-GFP and ChpT, a GTG start codon was used to moderate protein expression. In pCDF-duet (Invitrogen), *cckA* variants were cloned at site 1 bearing NcoI and NotI restriction site. These *cckA* variants were PCR amplified, including the T7 promoter, and cloned at NheI restriction site upstream of p15A ori in pACYC-duet. In a separate pACYC-duet plasmid, *cpdR-gfp-6xhis* was cloned at site 1.

***C. crescentus* vectors.** Plasmid pBXMCS2 was used for cloning *cckA* variants in between NdeI and EcoRI restriction sites. NdeI and KpnI restriction sites were used in pMCS5, pXMCS5 and pVMCS6 vectors[48]. pVMCS6 transformants were selected on the basis of exhibiting constitutive rather than vanillate-inducible GFP signal, indicating that expression was driven by the native promoter. pXMCS5 transformants were selected on the basis of xylose-inducibility. In the host strain, mChy-PopZ complements the severe Δ*popZ* mutant phenotypes in the *popZ::mChy-popZ* strain background[7].

### Bacterial cell culture
*E. coli* cells were grown at 30 °C overnight in Luria–Bertani (LB) liquid media or at 37 °C in LB agar plate supplemented with 1.5% agar. Liquid cultures were grown on rotor if not mentioned otherwise and all the growth of bacterial culture was measured by absorption $OD_{600}$. For *E. coli* strains, antibiotics were used at following concentrations: 50 μg ml$^{-1}$ ampicillin, 20 μg ml$^{-1}$ chloramphenicol, 50 μg ml$^{-1}$ spectinomycin, 12 μg ml$^{-1}$ oxytetracycline, 30 μg ml$^{-1}$ streptomycin, 30 μg ml$^{-1}$ kanamycin.

*C. crescentus* was grown at 28 °C in PYE liquid media or PYE agar plates supplemented with 1.5% agar. For antibiotic-resistant strains, antibiotics were used at the following concentrations: 2 μg ml$^{-1}$ chloramphenicol, 25 μg ml$^{-1}$ spectinomycin, 1 μg ml$^{-1}$ oxytetracycline, 5 μg ml$^{-1}$ streptomycin, 5 μg ml$^{-1}$ kanamycin. *C. crescentus* plasmids were transformed via electroporation or genes with associated antibiotic markers were transduced using phage φCr30. Prior to analysis, stationary phase cells were diluted 50× in fresh PYE and grown until they reached an $OD_{600}$ = 0.3. Xylose-inducible genes were induced by 0.2% final concentration of D-xylose for 2 h unless mentioned otherwise. *C. crescentus* synchronies were performed according to Toro et al.[49] but in PYE media at 28 °C using GB#228.

### Wide-field microscopy
Cells were immobilized on a 1% agarose gel pad and viewed with a Zeiss Axio Imager Z2 epifluorescence microscope equipped with a Hamamatsu Orca-Flash 4.0 sCMOS camera and a Plan-Apochromat 100×/1.46 Oil Ph3 objective. Zen 2 (Blue Edition) software was used for image capture and quantification. For fluorescence imaging, mChy was observed by excitation at 587 nm and emission at 610 nm, GFP by excitation at 488 nm and emission at 509 nm, and YFP by excitation at 508 nm and emission at 524 nm. Exposure times for imaging mChy,

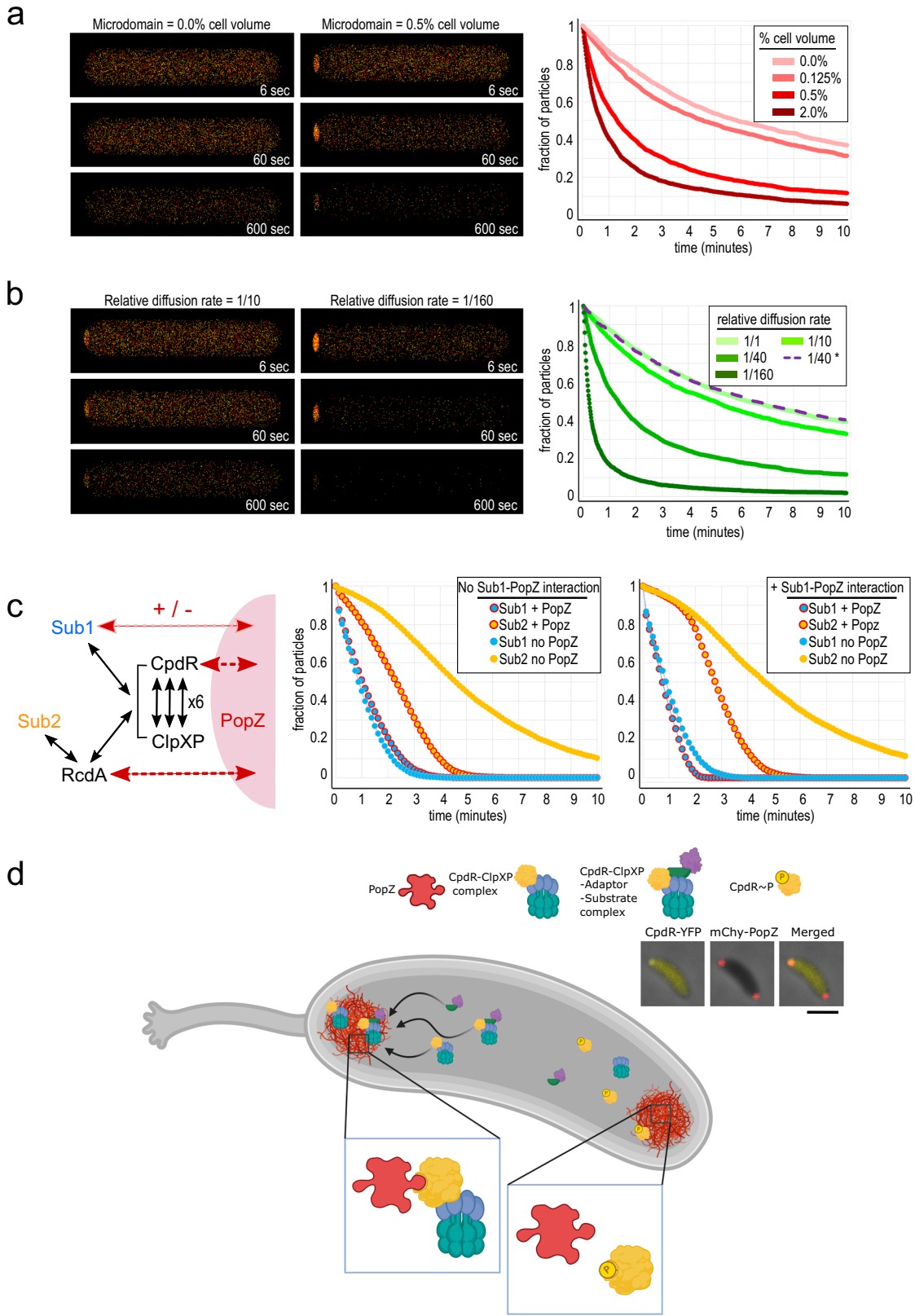

GFP and YFP-tagged proteins were 500 ms, 500 ms and 1000 ms, respectively, except that 50 ms exposures were used to image phase condensates. All microscopy was performed at ×1000 magnification.

For image quantitation, a cell was counted as having a polar focus if the appropriate fluorescence channel, when overlayed over the corresponding phase contrast image, displayed a clearly distinguishable increase in fluorescence intensity in the group of pixels in the area of the cell pole relative to the adjacent region of the cytoplasm. To quantify the fluorescence intensity of CpdR-YFP foci relative to cytoplasm, the spline tool in Zeiss Zen Blue software was used to manually outline the area of the polar fluorescence focus of CpdR-YFP and the remainder of the cell body in the YFP channel image, and the

**Fig. 6 | Conceptual models of substrate proteolysis in membraneless polar microdomains. a, b** Three-dimensional reaction-diffusion simulations with two types of particles, colored red and yellow, that disappear after colliding. Snapshots of cells from the indicated time during simulation are shown at left, and the number of particles remaining over time under different parameter conditions are shown at right. **a** Effect of varying the size of the polar microdomain while holding the particle diffusion rate at 1/40th the rate in bulk cytoplasm. **b** Effect of varying particle diffusion rates within polar microdomains while holding polar micro-domain size at 0.5% of total cell volume, or of limiting polar concentration to only one reactant (dotted line). **c** A model of *C. crescentus* proteolysis that includes low-

affinity interactions (black arrows) between substrates (Sub1 and Sub2), adaptors, and ClpXP protease, in which some proteins interact directly with PopZ (red arrows) and become concentrated in polar microdomains. Charts show the frac-tions of substrate particles remaining in simulations run with or without PopZ, either without (left) or with (right) a direct interaction between Sub1 and PopZ. **d** Localization of CpdR and associated ClpXP complexes as a consequence of asymmetric CckA signaling activity. Inset panels show fluorescence images of a *C. crescentus* stalked cell, where mChy-PopZ is localized to both poles and CpdR-YFP is localized to only the stalked pole. Scale bar = 10 μm. Source data are provided as a Source Data file. Graphics: BioRender.

circle tool was used to outline a neighboring region to measure local background fluorescence. After subtracting the background signal, the YFP signal of the polar focus was divided by the YFP signal of the whole cell ($n = 20$ in each of 2 biological replicates, with standard deviation showing differences between cells).

## Phos-tag gel assays
### Buffer composition
**Buffer A.** 0.5 M Tris HCl, 1.25 ml of 0.4% SDS (pH 6.8), 100 μl IgePal CA-630, 1.5 ml glycerol, up to 9.5 ml $H_2O$.

**Buffer B.** 920 μl Buffer A, 50 μl 2-mercaptoethanol, 10 μl EDTA free protease-phosphatase inhibitor cocktail (Halt™, Thermo Fisher), 20 μl lysozyme and 0.2 μl nuclease (CelLytic™ B plus kit, Sigma–Aldrich).

**Sample buffer 10X.** 0.5 M Tris HCl, 1.25 ml of 0.4% SDS (pH 6.8), 5 ml glycerol, 10 mg bromophenol blue, up to 10 ml $H_2O$.

**Cell lysis.** 1 ml cells of $OD_{600}$ 0.6–0.8 were centrifuged at $9000 \times g$ 2 min, washed 2× with LB or PYE and resuspended in 100 μl of freshly prepared Buffer B. After Incubating on a roller/rotor for 10 min in a 4 °C cold room, samples were centrifuged at $16,000 \times g$ for 5 min at 4 °C and the cleared supernatant was transferred to a new tube and kept on ice. After adding 10 μl of 10× sample buffer, 30 μl sample was loaded onto a prepared Phos-tag gel in a cold room at 4 °C.

**Phos-tag gels.** Ten to twelve percent gels were prepared according to manufacturer's instructions (Wako) with the exception of adding 40 μM $MnCl_2$ and 15 μM Phos-tag reagent as final concentration. The samples were run at 100 V in a 4 °C cold room until the dye front eluted from the gel. In-gel GFP or YFP fluorescence was imaged in a gel doc (Biorad Chemidoc™ MP) at 488 nm wavelength. Band intensities on gels were quantified using ImageJ 1.54 f.

## *E. coli* co-expression assay and fluorescence quantitation
Stationary phase *E. coli* cells bearing pACYC-duet and pBAD plasmids were diluted 100× in fresh LB and grown on rotor at 30 °C until an $OD_{600}$ of 0.3 (strain GB#1971–1977, Supplementary Table 1). Cells were induced with 0.2% arabinose for 3 h and 10 μM IPTG for the final 2 h to express mChy-PopZ and CpdR-GFP + ChpT + CckA, respectively. Image capture and quantification was performed as described by Nordyke et al.[19]. Cells with zero mChy-PopZ foci or foci at both poles were excluded from the analysis. Background-subtracted pixel intensities for each channel were measured along a straight line drawn lengthwise through mid-cell. Cubic spline interpolation was used to generate fluorescence intensity values for 100 equally spaced points along each line, then all points were normalized to 1 prior to averaging.

## Photobleaching
An Olympus IX-81 inverted confocal microscope equipped with a Yokogawa spinning disk (CSU X1) and a sCMOS camera (Orca- FLASH 4.0; Hamamatsu) was used for confocal microscopy. Excitation wave-lengths were controlled using an acousto-optical tunable filter (ILE4;

Spectral Applied Research) and a 405 nm FRAP laser was used with 100 mW nominal power. MetaMorph 7.7 software (MetaMorph Inc.) was used for image acquisition. FRAP time-lapse images were acquired using a 60×, 1.35 NA oil objective at 50 ms time intervals. To create bipolar mChy-PopZ containing cells, *E. coli* cultures were supple-mented with 30 μg ml⁻¹ cephalexin along with 0.1 mM IPTG and 0.2% arabinose at $OD_{600} = 0.3$, and incubated for 4 h at 30 °C. To plot Fluorescence Recovery After Photobleaching (FRAP) and Fluorescence Loss in Photobleaching (FLIP), time-lapse images of 20 different cells from each type were acquired. Fluorescence intensity of the photo-bleached area, unbleached area and background area were quantified using FIJI software for all the time points. Normalization of fluores-cence was performed by subtracting the background signal and mul-tiplying by the photobleaching coefficient. Photobleaching coefficients were obtained using a neighboring unbleached cell as a reference. To calculate the photobleaching coefficient for $T_n$ time point of the reference cell, the total cell fluorescence signal from the initial time point $T_1$ was divided by the total cell fluorescence signal from $T_n$.

## NMR protein–protein interactions
The expression and purification of ¹⁵N-enriched $PopZ^{\Delta134-177}$ protein was described previously[12,19]. For CpdR-GFP fusion protein and GFP expression, *E. coli* precultures were grown overnight from a single colony at 37 °C and 255 rotation-per-min (RPM) in LB with 30 μg/mL chloramphenicol or 100 μg/mL ampicillin, respectively. LB was inoculated with 0.4% (v/v) *E. coli* preculture, grown to mid-log phase, and then induced for 10 h at 37 °C and 255 RPM with isopropyl ß-D-1-thiogalactopyranoside (IPTG) for CpdR-GFP and with 0.2% (w/v) L-arabinose for GFP. Cells were harvested via centrifugation at $7000 \times g$ for 20 min at 4 °C and stored at −80 °C until further use. Cell pellets were resuspended in buffer containing 20 mM HEPES, 100 mM KCl, 2 mM $MgCl_2$, and 20 mM imidazole at pH 7.5, supplemented with Halt™ Protease Inhibitor Cocktail and Benzonase® Nuclease. After three passes through a French Press at 1750 PSI, cell debris was sepa-rated by centrifugation at $27,000 \times g$ for 30 min at 4 °C and the supernatant was passed through a 0.45 μm and then a 0.22 μm syringe filter. CpdR-GFP and GFP were purified from the filtered lysate by nickel affinity chromatography (HisTrap™ HP, 1 ml column) using an AKTA fast protein liquid chromatography system, using 20 mM HEPES, 100 mM KCl, 2 mM $MgCl_2$, and 20 mM imidazole as wash buffer and supplementing with 500 mM imidazole for elution. The eluate was dialyzed overnight into 50 mM $KH_2PO_4$, 20 mM NaCl, 5 mM EDTA at pH 6.5 for NMR experiments. Samples were concentrated using a 30 kDa molecular weight cutoff centrifugal filter (Amicon Ultra), and protein concentration was estimated via UV-Vis spectrophotometry. Sample purity was confirmed by 12% SDS-PAGE and visualization with Coomassie brilliant blue.

Transverse Relaxation Optimized Spectroscopy 2D ¹H-¹⁵N Het-eronuclear Single Quantum Correlation (2D ¹H-¹⁵N TROSY-HSQC) spectra were acquired using a 700 MHz Avance NEO NMR system equipped with a triple resonance helium-cooled cryoprobe. To char-acterize the binding of CpdR-GFP to $PopZ^{\Delta134-177}$, a series of samples

were prepared of 50 μM $^{15}$N-enriched PopZ$^{\Delta134-177}$ and natural abundance "unlabeled" CpdR-GFP fusion protein of varying concentrations (0, 125, 250, 375, 500, and 750 μM). The samples were supplemented with D$_2$O, NaN$_3$, and trimethylsilylpropanesulfonate to a final concentration of 5% (v/v), 4 mM, and 0.2 mM, respectively. Each 2D $^1$H-$^{15}$N TROSY-HSQC was acquired with 64 scans. The NMR datasets were processed using the NMRPipe software[50] and analyzed with the POKY software suite[51]. The PopZ$^{\Delta134-177}$ NMR spectra were assigned previously[19], thus amino acids were readily identified in the data. Combined $^1$H and $^{15}$N chemical shift perturbations (ΔHN) were calculated using the equation 1: $\Delta HN = \sqrt{(\Delta HN)^2 + (0.15 \times \Delta N)^2}$, where $\Delta H$ and $\Delta N$ are the chemical shift perturbations in ppm for $^1$H and $^{15}$N, respectively, and 0.15 is a scaling factor corresponding to the relative chemical shift dispersion in the $^1$H and $^{15}$N dimensions. Intensity perturbations were calculated as percentages of the 0 μM CpdR titration point peak intensities. Binding residues were determined by comparing the combined $\Delta HN$ chemical shifts and peak intensity perturbations for each residue. A chemical shift change larger than one standard deviation, corresponding to a continuous stretch of amino acids, was considered indicative of participation in binding.

### PopZ condensates and in vitro phosphorylation

To prepare proteins, PopZ, CpdR-GFP, CpdR$_{D51A}$-GFP were expressed using *E. coli* strains GB#169, GB#1969 and GB#1970. Cells were grown overnight to stationary phase, then diluted 100× in 1 L LB and grown back to OD$_{600}$ = 0.6 in a 37 °C shaking incubator at 200 RPM. After inducing with 1 mM IPTG for 6 h, cells were pelleted by centrifuging at 5000 × *g* for 30 min, resuspended in 20 ml HMK buffer (20 mM HEPES, 2 mM MgCl$_2$, 100 mM KCl pH 7.5) and lysed using a french pressure cell press (Sim-aminco) at 1000 psi pressure. Proteins were purified using HisPur cobalt resin (Thermo Scientific) according to the product literature batch purification method. Purity was assessed by SDS-PAGE and Coomassie blue staining, and yields were quantified by BCA protein assay kit (Thermo Scientific). Purified proteins were preserved in 100 μl aliquots at −80 °C after flash freezing in liquid nitrogen.

In vitro phosphorylation was performed with the indicated concentration of Acetyl phosphate (AcP) in HMK buffer with 20 mM MgCl$_2$ and at a final concentration of 100 μM CpdR-GFP. Samples were incubated at 37 °C for 6 h. To form PopZ condensates, 20 mM MgCl$_2$ was added to 50 μM purified PopZ for 10 min at room temperature. Next, PopZ condensates were mixed with 5 μM CpdR-GFP or GFP, and 7 μl of this mixture was transferred to a glass slide and covered with a coverslip. Imaging was performed immediately thereafter using phase contrast and the YFP fluorescence channels, with 50 ms exposure time, at ×1000 magnification.

Condensates of varying size were analyzed using Zeiss Zen Blue software. The linescan tool was used to draw a 10 μm line through the middle of each condensate, which was then separated into 160 bins of equal length. The fluorescence intensity of the bright ring of CpdR-GFP at the periphery of the condensate was obtained by taking the median value of the brightest 10 bins. The fluorescence intensity of the inner area of the condensates appeared to be affected by out-of-plane fluorescence, particularly in small condensates, and was therefore omitted from the analyses. The fluorescence intensity of the unbound pool of CpdR-GFP molecules was taken as the median value of the bins from outside the condensate. After subtracting background signal (observed in mounts without CpdR-GFP), we calculated the ratio of the fluorescence intensity of the bright peripheral ring of CpdR-GFP to the fluorescence intensity of the unbound CpdR-GFP molecules in the surrounding medium. To account for sample thickness, which altered fluorescence intensity from the out-of-focus pool above and below the

condensates, we used the formula $\frac{(R-(P-A))}{(P-(P-A))}$, Where $R$ is peripheral ring intensity, $P$ is the local unbound pool intensity, and $A$ is the average intensity of the unbound pool across all condensates in the sample set. The value for $A$ differed by less than 22% between samples.

### ClpXP substrate degradation (fluorescence and western blotting)

WT and *ΔpopZ C. crescentus* strains, expressing HA tagged CtrA RD + 15 (GB#1989, GB#1992), TacA (GB#1990, GB#1993), PdeA (GB#1991, GB#1994) were induced at OD$_{600}$ = 0.3 with D-xylose in 3 ml volume each for 2 h. Cells were washed 3 times by centrifugation and resuspension in fresh PYE, with final adjustment to ~3 ml at the same OD$_{600}$. Over the subsequent 2 h of growth on a rotary shaker at 28 °C, 300 μl sample aliquots were harvested and centrifuged at 9000 × *g* for 2 min. The cell pellets were resuspended in 30 μl SDS-PAGE sample buffer, flash frozen in liquid nitrogen, and preserved at −80 °C. Samples were loaded in a 10% SDS-PAGE gel and resolved at 100 V. After transferring to a PVDF membrane (0.45 μm pore size for HA-TacA and HA-PdeA samples, 0.22 μm pore size for HA-CtrA RD + 15 samples), Western blots were developed using an anti-HA mouse monoclonal primary antibody (Invitrogen 2-2.2.14) at 1:3000. Similar analyses were performed on strains expressing YFP-tagged CtrA RD + 15 (GB#1983, GB#1986), TacA (GB#1984, GB#1987), PdeA (GB#1985, GB#1988). Where chloramphenicol treatment was included, chloramphenicol was added at 30 μg ml$^{-1}$ after washing away the inducer, and cells were incubated for a further 5 min at 28 °C before initiating the time course. Band intensities on gels were quantified using ImageJ 1.54 f.

Fluorescence imaging was performed at 4 min time intervals. Fluorescence quantification was performed using FIJI plugin MicrobeJ 5.13i. We corrected the fluorescence values in *C. crescentus* time-lapse experiments to account for photobleaching. To do this, we induced YFP expression in *E. coli* cells from the pBAD promoter to a YFP fluorescence intensity level that was no more than 2-fold different than the *C. crescentus* cells we wished to analyze. Next, we imaged the *E. coli* cells under the same time-lapse conditions that were used to observe *C. crescentus*. The total fluorescence of 20 *E. coli* cells were observed over 20 consecutive time points and photobleaching coefficients were obtained by the method described in the previous section. Following background subtraction, to obtain fluorescence values for *C. crescentus* observations, YFP photobleaching coefficients corresponding to the appropriate time intervals were multiplied by the whole cell *C. crescentus* fluorescence values.

### Computational simulation

Smoldyn 2.71 simulations were run on a High Performance Computing cluster. Smoldith diffusion coefficients oyn files are provided in Supplementary Data 1, and a summary of the simulation parameters is provided in Supplementary Tables 3 and 4.

**Size scaling.** 1 size unit in Smoldyn equaled 10 nm in real space. Rod-shaped cells were created from cylinders and hemispheres, with length and volume that correspond to ~2.84 μm and 0.61 fL, respectively, consistent with measurements of stalked cells[52]. The volume of polar PopZ compartments has been measured by 3D super-resolution fluorescence imaging[53]. A mid-range value of $3.4 \times 10^6$ nm$^3$ was chosen for standard cell models, which corresponds to ~0.56% of the total cytoplasmic volume.

**Time scale and diffusion rates.** The parameters for diffusion coefficients are based on single-particle tracking studies in *C. crescentus* cells[25]. ChpT and CtrA were found to move sub-diffusively in bulk cytoplasm, with diffusion coefficients of 1.8 μm$^2$/sec, which translates

to 18 units$^2$/ms in the simulations. ChpT and CtrA were observed to continue to exhibit Brownian diffusion while co-localized with PopZ at cell poles, but at substantially slower rates of 0.1 μm$^2$/s and 0.01 μm$^2$/s, respectively (corresponding to 1 units$^2$/ms and 0.1 units$^2$/ms). In simulated cells, the diffusion coefficient of all particles in bulk cytoplasm was set to 20 units$^2$/ms, and the standard rate for particles that interact with PopZ in polar microdomains was set to 0.5 units$^2$/ms. For all simulations, the time step was 0.1 ms, which corresponds to a resolution of ~20 nm[40].

**Number of particles.** In simple models with only two types of particles, the number of each was set to 2500. For the more complex model, global measurements of gene transcription and translation rates in *C. crescentus* cells provided an accurate estimate of the total number of each protein[24]. To approximate the number of proteins following cell division, the total amount of protein produced per cell in PYE media was divided by two and rounded to the nearest hundred. Thus, the number of ClpXP hexamers was set at 1000, CpdR at 7000, and RcdA at 2200. Since many different proteins are degraded by ClpXP and altogether there are thousands of individual substrates, the number of CpdR-dependent and RcdA plus CpdR-dependent substrates was arbitrarily set at 8,000 each.

**Reaction rates and binding kinetics.** Smoldyn represents proteins as point-like particles, and updates each particle's position over iterative time steps, assigning a random direction and a distance calculated from its diffusion coefficient. Interactions between particles are determined by proximity. If two reactive particles are positioned within a defined binding radius at the end of a time step, those particles will interact. If they remain associated for a time, they will diffuse as a combined particle and the timing of separation is defined by an off-rate. Reaction rates and binding equilibria are therefore determined by the combined influences of binding radii, particle concentrations, diffusion rates, and off-rates. For the simple two-particle model, an empirical process was used to determine a binding radius of 0.025, which yielded a reaction rate that roughly matched the observed rate of protein degradation in *C. crescentus* cells (Fig. 4b). For the complex model, Smoldyn calculated the binding radii from user-defined values for on-rate and off-rates. We used the same values for all particle interactions ($1 \times 10^4 \, M^{-1} s^{-1}$/0.166 px$^3$ms$^{-1}$ for on-rate and 10 s$^{-1}$/0.01 ms$^{-1}$ for off-rate), which are well within the range of measured rate constants for dimeric complexes[54] and correspond to a weak association with a $K_D$ of 100 micromolar. All proteins were treated as monomers except for ClpXP, which was modeled as a hexamer that could bind up to six CpdR and/or CpdR-associated particles. With these parameters, the binding radius was 1.6 nm, which is less than the radius of the proteins being modeled, meaning that that they will not interact every time they collide. This could reflect conformational and geometric constraints on productive collisions.

### Reporting summary

Further information on research design is available in the Nature Portfolio Reporting Summary linked to this article.

## Data availability

The datasets generated and analyzed during the current study are available in the source data file, or, in the case of microscope images of experimental replicates, available from the corresponding author upon request. Source data are provided with this paper.

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

## Acknowledgements

The authors would like to thank Steve Andrews for comments and advice, and Jack C. Sylvester for technical assistance. K.V. was supported by the National Science Foundation Major Research Instrumentation Award (DBI-1828319) and an Institutional Development Award (IDeA) [CIBBR, P20GM113131] from the National Institute of General Medical Sciences of the National Institutes of Health. G.B. was supported by the National Institutes of Health under award numbers 2P20GM103432 and R01GM118792, and by the National Science Foundation under award number 2225849.

## Author contributions

Author contributions: Y.M.A., L.M.B., K.V., and G.R.B. designed research and analyzed data; Y.M.A., L.M.B., and G.R.B. performed research; Y.M.A., K.V., and G.R.B. wrote the manuscript.

## Competing interests

The authors declare no competing interests.
