## [Transparent Peer Review file · Nature Communications]

Phospho-signaling couples polar asymmetry and proteolysis within a membraneless microdomain in *Caulobacter crescentus*.

Corresponding Author: Dr Grant Bowman

Version 0:

Reviewer comments:

Reviewer #1

(Remarks to the Author)

In this manuscript, the authors use a combination of live cell imaging, in vitro experiments, and simulations to highlight the importance of the polar assembly hub PopZ in regulating polar asymmetry and proteolysis. They show that phosphorylation of CpdR regulates its interactions with PopZ and validate the importance of CckA in this regulation. In addition, they propose that the PopZ microdomain can locally concentrate protease, adaptors and substrates which should increase degradation rates, linking the polar recruitment of these factors to cell cycle proteolysis. Finally, they use reaction modeling software with parameters derived experimentally or empirically that PopZ condensates can dynamically increase concentrations of protease, adaptors and substrates to stimulate degradation. Overall, the conceptual framework of the paper is strong and the conclusions are reasonable. There is a need for additional quantification of results, to explore the simulation work more, to clarify some experimental details, and to describe alternate interpretations of their models.

Detailed comments:

1. For all the microscopy images where co-localization of proteins are described, it is important to support the claims with quantification of numbers and replicates. This is particularly critical as most of the conclusions made in the paper draw from these currently qualitative results. For example, in line 88 the statement is made that "When we expressed a hyperactive-kinase (H+) form of CckA (G319E), the ratio of phosphorylated to unphosphorylated CpdR-YFP was relatively high" and it is challenging to see this just with the representative images. Including some quantification of co-localization, e.g, number of cells with co-localized foci over total number of fluorescent cells, and the number of biological replicates will add strength to these arguments.

2. In Figure 2d, the PopZ condensates clearly recruit CpdR-YFP, but the effect of phosphorylation is more challenging to interpret as there are many puncta in the CpdR-GFP +AcP samples. How are these heterogeneous puncta incorporated into the overall fluorescent signal reported? Based on the methods, it seems that 10 condensates of 'similar sizes' were measured for this assay; however, measuring fluorescence/area should make it possible to use a range of sizes and shapes so that more quantitative treatments can be made.

3. In the same subfigure, why does phosphorylation of CpdR-YFP result in puncta? Do the number or intensity of puncta scale with phosphorylation? This line of inquiry addresses how to interpret the model that PopZ condensates recruit CpdR only when dephosphorylated as if phosphorylation of CpdR causes oligomerization or aggregation, then that would explain why it no longer binds PopZ.

4. In Figure 2f, a larger range of phosphorylation would strengthen the point that CpdR phosphorylation is negatively correlated with PopZ partitioning. The lowest concentration of AcP yields 60% CpdR~P, but it seems that lower concentrations (or addition of some phosphatase) could allow for accessing the 0-60% range.

5. In Figure 2g, CpdR dissociates from PopZ with a half-time of around 0.5 seconds. It would be good for the authors to address how this short timescale affects the in vitro measurements of Figure 2d. For example, the imaging time for YFP (based on the methods) is 1000 ms - does this result in a gradual loss of signal over time consistent with the measured half-life?

6. For Figure 3. Are the fluorescently tagged constructs used in this study the only allele of these genes in the strain, or are

they merodiploid strains? If they are the only alleles are there any physiological defects in these strains as tagging proteins can often lead to some loss of function. Based on the strain table and plasmids, it appears that the vanillate inducible plasmid is being used for the integration which would result in a second inducible, but possibly basally expressed, copy of the gene as well.

7. In Figure 4b, it is not clear what the blue and orange histograms represent. Are they the polar localization of stalked versus swarmer? Or are they the fraction of cells with polar localization at the time points shown (in which case, why is it a smooth distribution over time?). More clarity is needed for this particular subfigure.

8. In Figure 4, the *in vivo* results showing rapid degradation of PdeA, TacA and CtrA constructs is very strong and the authors should be complemented for the clarity of the results. It is equally striking that there is still substantial degradation seen in Δ popZ strains based on the quantified data. Did the authors look at this degradation by microscopy as well? Based on their model, you might imagine there to be no cell division dependence to this turnover.

9. In lines 180-185 the authors raise an important concern with using translation inhibitors to evaluate protein degradation. The possibility that chlor addition affects cell cycle progression (and therefore protein stability) is noteworthy. It would be excellent to complement this observation with microscopy experiments showing this same halting of degradation in single cells as well.

10. The authors should note the confounding factor of the xylose-based reporter system they are using with respect to cell-cycle regulated expression (such as that for CtrA and TacA). Degradation is controlled during the cell-cycle such that the most robust activity often correlates with the lowest expression of the targets (such as CtrA and TacA). By using a non cell-cycle dependent promoter, the authors may be seeing effects on levels of proteins that is not seen with the physiologically related proteins. Looking at endogenous levels of CtrA and TacA (for example) would resolve this - however, it is not critical to address this experimentally, but it should be commented on in the results.

11. The simulation results are consistent with the experimentally derived model that dynamic concentration of reactants yields rapid degradation. However, there are a few specifics that should be addressed. There should be more details about the assumptions in the simulation - it is mentioned that some parameters are empirically derived, some experimentally determined. The empirically derived binding radius is 0.025 units, which seems to correspond to 2.5 Å based on the conversions described, but this seems quite small if we are considering the center of particles for these proteins. Is there any biological significance to this value, what is the effect of adjusting this binding radius? Does it affect the diffusion measurements/assumptions? Some explanations here would be excellent.

12. Along the lines of this simulation, the notion that concentrating reactants to induce faster product formation (in this case degradation) is not altogether surprising. It would be ideal to address the specific use case of the CpdR, RcdA, and PopA dependent degradation that is being experimentally probed. Specifically, can the simulation predict the order of degradation rates for the different substrates that are observed experimentally and referred to in lines 170-174?

line 167: typo " In stalked cell progeny, all three proteolysis substrates were cleared from the cell between before cell separation"

Reviewer #2

(Remarks to the Author)

In the study titled "Phospho-signaling couples polar asymmetry and proteolysis within a membraneless microdomain in *C. crescentus*", the authors explore the relationship between cell-cycle regulated proteolysis and the polar PopZ condensate in *Caulobacter crescentus*. Three pivotal observations are drawn from this investigation. The main findings of this work are that (i) the phosphorylation state of the protein CpdR, a cell cycle-regulated adaptor protein, affects its localization with the PopZ microdomain, (ii) CpdR further recruits and concentrates the associated members of the ClpXP proteolysis complex into the PopZ microdomain at the poles for timely degradation, (iii) discussion on how localization to PopZ regulates rates of degradation.

Points (i) and (ii) echo some of the previous findings in the field. The authors should consider executing additional experiments to justify their new findings relating to these two points, as listed below. Moreover, critical information that would allow us to assess the work relating to point (iii) is missing.

Below are the points that should be addressed:

Comments related to Figure 1:

1. Lack of evidence for a direct relationship between CpdR phosphorylation state and PopZ: The title of Figure 1, "CpdR phosphorylation state influences its co-localization with PopZ," suggests a direct relationship between the phosphorylation state of CpdR and its co-localization with PopZ. The data presented illustrate that the polar localization of CpdR is cell-cycle regulated (shown before, Iniesta et al., 2006), that the unphosphorylated CpdR localizes to both poles (shown before, Iniesta et al., 2006), and that the hyper-phosphorylated version doesn't localize to poles (new data). This indicates a correlation between polar localization and phosphorylation state but not necessarily causation with PopZ. Please address this point. In addition, the definition of "polar localized" is not clear. How much enrichment of fluorescence signal at the pole accounts for

polar localization? Finally, a quantitative analysis of multiple cells in Figures 1e and 1g would offer more clarity.

2. Clarification on *cckA* Mutants: From my understanding, the *cckA* mutants (H+ and K-) in this study are expressed using an inducible plasmid while the native *cckA* gene is still present and functional (in strains GB#1980 and GB#1981). This dual perturbation—abundance and kinase activity—should be elucidated in the main text. To better distinguish these effects, perhaps the authors could include a "CckA WT on plasmid" in 1f. I'm assuming "CckA WT" currently denotes a chromosomal copy of *cckA*. This additional strain would use the same plasmid backbone as H+ and K- but express the WT *cckA* gene.

3. CckA and CpdR Relationship: Given that CckA functions as a kinase for CpdR, the observed increase in phosphorylated CpdR in the H+ mutant, shown in 1f, isn't surprising. However, I find it perplexing that the CckA WT shows a profile close to identical to CckA K-. Moreover, why does there appear to be increased unphosphorylated CpdR_D51A in the K- context? A side note: the letter 'K' is absent in the rightmost lane of 1f.

4. PopZ Localization in Different Backgrounds: In the H+ context depicted in 1e, PopZ's localization seems confined to a single pole rather than two in dividing *C. crescentus* cells. Is that the case? And if so, how can that be explained? Please add a quantification of PopZ's localization across the varied CckA mutant contexts.

5. Phosphogel Replication & Quantification: In relation to the phosphogels presented in Figure 1 (and similarly in Figures 2 and 3), please include three biological replicates. Additionally, please quantify these gel band intensities using standard deviations to improve data robustness.

Comments related to Figure 2:

6. PopZ / CpdR Interaction: This work very much relies on the assumption that CpdR directly binds PopZ. While there is data to indirectly support this assumption, the authors do not provide direct evidence. The authors show co-localization imaging assays in *E. coli* and integration of CpdR molecules into PopZ condensates *in vitro*. As the manuscript's central claim hinges on CpdR's direct binding with PopZ, this must be definitively established *in vitro* using one of the well-established protein binding assays. It's worth noting that several alternate mechanisms could result in the enrichment of CpdR within PopZ clusters at the cellular pole, especially when the observed enrichment is relatively modest.

7. Condensate images: Concerning the condensate images presented in image 2d, it is difficult to assess the condensate behavior. Please show a comprehensive figure that encompasses multiple condensates for each experimental condition. Alongside this, a detailed quantitative analysis detailing the extent of CpdR's incorporation within these condensates would be beneficial.

8. The authors suggest in line 122: "In conjunction with the *E. coli* reconstitution experiments, these findings imply that CpdR phosphorylation may hinder its direct interaction with PopZ." While this is an intriguing interpretation of the presented data, it remains speculative without additional experimental corroboration.

9. Interpretation of Figure 2b, like many of the other imaging data, would benefit from quantitative analysis of many cells. Certain instances display a CpdR focus without a concurrent PopZ focus, as observed in the top panel with WT CckA and CckA K-. This leads to the following queries: (i) Can CpdR foci be discerned even when PopZ is not prominently visible or does the mere presence of a minimal PopZ focus suffice to localize CpdR? (ii) Is it possible for CckA to play a role in CpdR polar localization that is independent of PopZ?

Comments relating to Figure 3

10. Please add quantification across many cells for the co-localization data shown in Figures 3c and 3d. It is very difficult to interpret the data as presented.

11. Some of the data presented as part of this figure was already shown, including that CpdR controls ClpXP localization to the pole (Iniesta et al. 2006) and that CpdRD51A leads to bi-polar localization of ClpXP (Iniesta et al. 2006). The new data here is that hyper-phosphorylated alters RcdA and ClpXP localization. However, more quantitation is required to support these claims, as indicated in point (10).

12. In addition, Nordyke et al. 2020 showed by NMR that RcdA directly binds PopZ. Can the authors explain why and how CpdR phosphorylation state seems to affect RcdA recruitment to PopZ?

Comments relating to Figure 4

13. In Fig. 4b, the authors have shown that the polar localization of three proteolysis substrates, PdeA, TacA, and CtrA-RD+15, governs their time of degradation in the swarmer and stalked cell progeny. Please explain why the polar localization of the substrates differs between the swarmer and stalked cells. What are the factors that contribute to that? Also, what exactly drives the proteolysis in the swarmer cell where CpdR does not localize in the PopZ microdomain?

14. Please indicate that the three substrates (PdeA, TacA, and CtrA-RD+15) are all expressed on inducible plasmids. This is not clear in the main text and is confusing to the reader.

15. The discussion in lines 180-185 is unclear. The authors write, "... we blocked new protein synthesis by treating cells with chloramphenicol. Over subsequent time points, a subpopulation of cells in wildtype cultures retained large quantiles of substrates for more than 60 minutes, but this was not observed in Δ popZ cultures (Extended Data Fig. 3a)." That means that protein degradation works better in Δ popZ cells in this experiment. They then write: "We propose that chloramphenicol treatment interferes with the measurement of degradation rates by preventing cell cycle progression in swarmer cells, locking them into a stage of high substrate stability. Δ popZ strains, whose cell division is often uncoupled from the cell cycle, do not appear to produce significant numbers of this cell type." There is no evidence in this paper to substantiate these claims, and other mechanisms are likely possible. This data is confusing, and the authors should consider either removing it or providing additional experiments to explain these results.

16. In lines 187-188, the authors write, "Under these conditions, ClpXP substrates were degraded up to twice the rate in wildtype cells compared to Δ popZ." Looking at Figure 4c, this does not seem to be the case. TacA and PdeA show modest to no effect in clearance in WT vs. Δ popZ cells, at least in the blots presented. For the case of CtrA, we know that its degradation is phosphorylation-state dependent. Can the disruption of CtrA dephosphorylation affect the slower rate of its degradation? Also, the authors say that the quantitation is based on three biological replicates. Please show these

replicates in SI.

Comments relating to Figure 5

17. The authors have used simulations to show how increasing the concentration of substrates within the PopZ microdomain enhances the rate of proteolysis. Many details of binding rates are missing, to the point that it is difficult to assess the quality of this section.

18. In extended data controls, I would also like to see that the simulations recapitulate the data presented in Figures 1-3. For example, CpdR, ClpXP, and RcdA localization in the different mutational backgrounds.

Other comments:

19. In Fig. 1b, the authors should provide error bars as it will give a better idea of the variation in the data.

20. In Fig. 2e, the authors should replace the gel image with a cleaner version to better understand the bands.

21. The majority of figures in the extended data figures lack scale bars.

22. Significant proofreading of the manuscript is required.

Reviewer #3

(Remarks to the Author)

In this manuscript, the authors characterise the dynamic localisation of polar proteins in the bacterial model *Caulobacter crescentus*. The authors were interested in understanding how proteins, such as the response regulator CpdR, are localised at one pole while interacting with the bipolar hub PopZ. In particular, they show data strongly suggesting that only unphosphorylated CpdR interacts with PopZ. Based on that, they propose that the PopZ:CpdR interaction is driven by the asymmetric activity of the the bipolarly localised histidine kinase CckA. Indeed, CckA was shown to work as a kinase at the "new" pole and as a phosphatase at the "old" pole, thereby increasing the local concentration of unphosphorylated CpdR at the old pole, that is where it interacts with PopZ. On the other hand, they show that this PopZ-dependent unipolar localisation of CpdR enhances rapid ClpXP-dependent proteolytic degradation of substrates at the old pole. The question raised in this work is original and important. Although the data support the conclusions, then manuscript could be improved (see below).

Major comments

- First, I don't understand why the authors only quantified the localisation of CpdR-YFP (Figure 1b). I would suggest to quantify all the localisation data (Fig 1e, 1g, 2b, 3c, 3d) instead of only showing a few cells. This is particularly critical when differences are tiny. For instance, the authors state "When we expressed a kinase-deficient (K-) form of CckA (H322A), the ratio was substantially lower, and cells exhibited robust polar foci." Terms like substantially or robust do not indicate to what extent the effect is observed.

All the phos-tag gels should also be quantified (Fig 1c, 1f, 2c) to confirm the effects expected from the mutants used. For example, I don't see a substantial change in the phosphorylated/unphosphorylated ratio of CpdR in the K- mutant of CckA on figure 1f.

- I would also suggest the authors to strengthen their evidence of a direct interaction between unphosphorylated CpdR and PopZ. First, they should quantify the fluorescence data shown in figure 2d.

Then, they could purify native CpdR (WT or point mutants), instead of CpdR-YFP derivatives, and perform in vitro assays which allows measurements of affinity constants between purified PopZ and CpdR variants. For example, ITC was already used to demonstrate direct interaction between PopZ and ZitP (Bergé et al., 2016 DOI: 10.7554/eLife.20640).

Maybe a phosphomimetic mutant of CpdR (D51E) could also be constructed and characterised? Such a mutant should display opposite phenotypes and behaviours to the D51A mutant.

- The authors suggest that CpdR may influence RcdA localisation indirectly, via ClpX. To check that, RcdA-GFP could be localised in a *cpdRD51A ΔsocAB ΔclpX*. Alternatively, *RcdA_ΔC*, which cannot interact with ClpX::CpdR (Joshi et al., 2015 <http://dx.doi.org/10.1016/j.cell.2015.09.030>)

Minor comments

- The authors should explain how the strains were constructed and designed. For example, how could a CckA kinase-deficient strain survive knowing that *cckA* is essential? Some important details are missing

- Lines 183-185: the interpretation of the results shown in Extended Data Fig. 3a is hard to understand, that is chloramphenicol interfere with degradation rates by preventing cell cycle progression. It looks like rather an hypothesis than an argument.

Is there a way to check if this hypothesis is correct?

- Line 74: phosphorylated → phosphorylated

- Line 79: unphosphorylated → unphosphorylated

Reviewer #4

(Remarks to the Author)

Version 1:

Reviewer comments:

Reviewer #1

(Remarks to the Author)

The authors have addressed all of my major comments and the revised paper is excellent. The attention to the responses and additional data are appreciated.

Reviewer #2

(Remarks to the Author)

The authors have effectively addressed the comments raised by the reviewers. However, one major issue must be resolved before the paper can be considered for publication: the in vitro data shown in Figure 3b-d. The authors used partitioning of CpdR-GFP into PopZ condensates to determine whether CpdR interacts with PopZ. This strategy assumes that enrichment of CpdR within the condensate indicates binding. Although valid, the implementation used in this work has issues. The authors measured the enrichment of CpdR mutants by comparing the average total GFP intensity within a PopZ droplet to that of CpdR WT. This metric is not standard practice in the field, which favors partition coefficient measurement (examples include PMID: 33169001, PMID: 33395293, PMID: 35245500).

There are several problems with this method.

First, the distribution of CpdR inside PopZ condensates, as indicated by the six condensates presented, is not straightforward. In two cases, CpdR appears to be uniformly distributed within the condensate, while in four cases, it decorates either the full or partial peripheral ring around the entire droplet. In the cases where the signal is only on the peripheral ring (which would still indicate binding), is the average taken on the whole condensate or only the outer ring?

Second, for phosphorylated CpdR (CpdR-GFP+AcP), accumulation is clearly seen on the peripheral ring of the condensate. The signal is less bright, which could be due to lower affinity, different imaging settings, or inherent differences in PopZ condensates in response to AcP. I suspect that using a partition coefficient measurement, which quantifies the fraction of phosphorylated CpdR within condensates, would show low-affinity binding compared to the current approach, which considers relative changes in fluorescence intensity.

Finally, in this dataset, the external area of the condensate is consistently black, including in the GFP control, which would be expected to show a uniform yellow signal throughout.

For these reasons, the data presented in Figure 3b-d does not conclusively support the conclusion that CpdR binds PopZ only in its phosphorylated state.

I recommend reanalyzing the data using the partition coefficient method, with at least 50 droplets per condition, including two additional biological replicates. Additionally, depositing the droplet images (and all other images) to a platform like Zenodo or Dropbox would enhance transparency. An orthogonal approach to demonstrate binding specificity upon phosphorylation could be explored using NMR.

Minor comment: y-axis labeling is missing for Fig. 3d, and Fig. 3b is not referred to in the main text.

Reviewer #3

(Remarks to the Author)

All the points highlighted by the reviewers have been properly addressed and new data requested by the reviewers have been incorporated in the revised manuscript, further supporting the main conclusions. In the present form, I do not have any other concern or request for this ms.

Version 2:

Reviewer comments:

Reviewer #2

(Remarks to the Author)

Thank you for including the calculation of the partition coefficient. The partition coefficient is a straightforward concept: it is the ratio of fluorescence intensity inside the condensate to that outside the condensate. Typically, this calculation involves:

(average fluorescence intensity inside the condensate minus background) divided by (average fluorescence intensity outside the condensate minus background)

In this manuscript, the authors used a variation of this method by focusing specifically on the bright peripheral ring versus the intensity outside the droplet. While this approach is unconventional and could introduce variability (due to differences in ring thickness across condensates), it's reasonable to keep it as is for the following reasoning:

The authors define partition coefficient as:

$$(R-(P-A))/(P-(P-A))$$

Where:

- R is the signal from the ring,
- P is the signal around the specific condensate,
- A is the average signal around all condensates.

Simplifying this, the calculation becomes:

$$(R-P+A)/A$$

If P is close to A, as indicated by the authors, this simplifies to R/A, which is close to the common practice.

Regardless, the final quantification compares fold changes. CdpR-GFP+AcP condition shows a fold change greater than 1, indicating some recruitment to the condensate. This is particularly evident as the "R" value is calculated from pixels that are part of the peripheral ring, which is still detected in the CdpR-GFP+AcP condition. Therefore, I recommend adjusting the phrasing in the main text.

In lines 155-159, the current statement is:

"Next, we phosphorylated CpdR-GFP by pre-incubating it with acetyl phosphate (AcP), using CpdRD51A-GFP as a non-phosphorylatable receiver to control for the presence of AcP (Fig. 3c). We found that CpdR-GFP phosphorylation inhibited the partitioning of CpdR-GFP into PopZ condensates, while the same treatment had no such inhibitory effect on the CpdRD51A-GFP control, nor did it affect protein mobility in Phos-tag gel electrophoresis (Fig. 3d). These results suggest that CpdR phosphorylation inhibits its physical interaction with PopZ."

Please revise the phrase "inhibited the partitioning of CpdR-GFP into PopZ condensates" to "reduced the partitioning of CpdR-GFP into PopZ condensates" for better clarity and accuracy.

Additionally, the reference to Fig. 3d in this section should refer to Fig. 3c.

Regarding Figure 3d, I recommend replacing the bar plot with either a scatter plot with box or violin plot that includes the original data points. Also, please include the raw data in Excel format to enhance transparency and reproducibility.

Lastly, please update the legend for Supplementary Figure 1 to indicate that Supplementary Figure 1c refers to Fig. 3c (not Fig. 3b) for consistency.

Brown text is part of the final response to review.

Blue text refers to revised text or other changes that must be copied / incorporated into the final revised manuscript.

REVIEWER COMMENTS

Reviewer #1 (Remarks to the Author):

In this manuscript, the authors use a combination of live cell imaging, in vitro experiments, and simulations to highlight the importance of the polar assembly hub PopZ in regulating polar asymmetry and proteolysis. They show that phosphorylation of CpdR regulates its interactions with PopZ and validate the importance of CckA in this regulation. In addition, they propose that the PopZ microdomain can locally concentrate protease, adaptors and substrates which should increase degradation rates, linking the polar recruitment of these factors to cell cycle proteolysis. Finally, they use reaction modeling software with parameters derived experimentally or empirically that PopZ condensates can dynamically increase concentrations of protease, adaptors and substrates to stimulate degradation. Overall, the conceptual framework of the paper is strong and the conclusions are reasonable. There is a need for additional quantification of results, to explore the simulation work more, to clarify some experimental details, and to describe alternate interpretations of their models.

Detailed comments:

1. For all the microscopy images where co-localization of proteins are described, it is important to support the claims with quantification of numbers and replicates. This is particularly critical as most of the conclusions made in the paper draw from these currently qualitative results. For example, in line 88 the statement is made that "When we expressed a hyperactive-kinase (H+) form of CckA (G319E), the ratio of phosphorylated to unphosphorylated CpdR-YFP was relatively high" and it is challenging to see this just with the representative images. Including some quantification of co-localization, e.g, number of cells with co-localized foci over total number of fluorescent cells, and the number of biological replicates will add strength to these arguments.

We have added two new figure panels with quantitative information that describes the results of our fluorescence microscopy of live cells. New Figure 1h provides quantitative analysis of the *C. crescentus* strains described in 1d-g. New panel 4d provides quantitative analysis of the *C. crescentus* strain described in 4b-c.

The revised results section also provides an explanation for one strain that was not quantified in 4d. The revised text states "In a $\Delta cpdR$ background, RcdA-GFP was mostly diffuse, but appeared to have residual polar foci that were difficult to reliably discern and quantify." "In a $\Delta cpdR$ background,

RcdA-GFP was almost completely diffuse, although many cells retained faint polar foci that were difficult to discern and quantify (Supplementary Fig. 4a)."

We provide a magnified image of these faint foci in new supplementary data Figure 4a. For these and other panels describing quantitative analysis of cell images (1b, 2c, S2b), we also provide information on the number of experimental replicates and the number of cells quantified per replicate in the associated figure legend.

2. In Figure 2d, the PopZ condensates clearly recruit CpdR-YFP, but the effect of phosphorylation is more challenging to interpret as there are many puncta in the CpdR-GFP +AcP samples. How are these heterogeneous puncta incorporated into the overall fluorescent signal reported? Based on the methods, it seems that 10 condensates of 'similar sizes' were measured for this assay; however, measuring fluorescence/area should make it possible to use a range of sizes and shapes so that more quantitative treatments can be made.

We provide a more detailed explanation of our quantitative analyses of droplet fluorescence in the revised figure legend. We have also increased the numbers of droplets that we analyzed (now over 20 in each condition across three separate trials) and taken data from a wider range of droplet sizes, with representative examples shown in new images in panel 3a. The revised figure legend now states:

"c, Average fluorescence intensities of condensates in a, normalized to the value obtained for CpdR-GFP. $n > 10$ per replicate; bar = standard deviation from 3 replicates; brackets indicate P-values from two-sided unpaired t-test: ** = 0.005, ns=non-significant."

3. In the same subfigure, why does phosphorylation of CpdR-YFP result in puncta? Do the number or intensity of puncta scale with phosphorylation? This line of inquiry addresses how to interpret the model that PopZ condensates recruit CpdR only when dephosphorylated as if phosphorylation of CpdR causes oligomerization or aggregation, then that would explain why it no longer binds PopZ.

We tracked the fluorescence heterogeneity in droplets across trials and different conditions (with and without AcP, and with and without CpdR phosphorylation). We found that fluorescence pattern heterogeneity could be apparent as either peripheral puncta on droplet edges (shown for the +AcP CpdR samples in the originally submitted manuscript) or as a peripheral ring around the whole droplet (shown in several droplets in the revised image panels). We found that the edge effects tended to be more pronounced if the droplets had a longer time to associate with glass surfaces in our incubation chambers, and therefore limited our total incubation time to less than fifteen minutes, as described in our

methods section. Since the edge effects arise from some qualities of the *in vitro* system and incubation conditions, we chose not to comment on their relevance in the manuscript text. Notably, we haven't ever observed brightened peripheral rings at the edges of PopZ microdomains in *E. coli* or *Caulobacter* cells.

4. In Figure 2f, a larger range of phosphorylation would strengthen the point that CpdR phosphorylation is negatively correlated with PopZ partitioning. The lowest concentration of AcP yields 60% CpdR~P, but it seems that lower concentrations (or addition of some phosphatase) could allow for accessing the 0-60% range.

We have removed the data on intermediate levels of CpdR-GFP phosphorylation in the revised manuscript. The original panel gave the impression that we claimed a linear correlation between the percent of CpdR-GFP phosphorylation and the partitioning of CpdR-GFP into droplets, and this might have been extrapolated to imply a lack of cooperativity in binding. However, given the inconsistency in our ability to manage the levels of CpdR-GFP phosphorylation *in vitro*, we decided that there were not enough data points to provide a confident trend line or any conclusions about binding cooperativity or the lack thereof, and that the simpler +/- phosphorylation experiments presented in the revised figure provide more clarity.

5. In Figure 2g, CpdR dissociates from PopZ with a half-time of around 0.5 seconds. It would be good for the authors to address how this short timescale affects the *in vitro* measurements of Figure 2d. For example, the imaging time for YFP (based on the methods) is 1000 ms - does this result in a gradual loss of signal over time consistent with the measured half-life?

Overexposure of our fluorescence images of *in vitro* condensates (now shown in Figure 3a) would show that they are surrounded by a large and likely inexhaustible pool of free CpdR-GFP in the surrounding buffer. Longer incubation times would result in higher average fluorescence intensity and also greater edge fluorescence intensity heterogeneity (as explained in response to comment #4). This suggests that the partitioning shown in 3a has not yet reached a steady state, but since the edge effects are changing for reasons we don't understand, we decided that it was better to take images at early time points.

Also, we mistakenly omitted the YFP channel exposure time for these experiments in the original manuscript. The revised methods section now indicates the image capture exposure time for these experiments, which was 50 msec.

6. For Figure 3. Are the fluorescently tagged constructs used in this study the only allele of these genes in the strain, or are they merodiploid strains? If they are the only alleles are there any physiological defects in these strains as tagging proteins can often lead to some loss of function. Based on the strain table and plasmids, it appears that the vanillate inducible plasmid is being used for the integration which would result in a second inducible, but possibly basally expressed, copy of the gene as well.

In strains presented in Figure 4 of the revised manuscript (Figure 3 in the original submission), RcdA-GFP and ClpX-GFP were introduced by single-copy genomic integration, and expressed under control of a vanillate-inducible promoter. Here, we modified the strains with the genomically integrating vector pVMCS-6, where the P_{van} sequence is functional but only 126 base pairs in length, and therefore too short to be used as a site for genomic integration. Consequently, this greatly favors integration at the *rcdA* or *clpX* locus, which would place *rcdA-gfp* or *clpX-gfp* downstream of the native promoter, and leave an untagged copy of *rcdA* or *clpX* downstream of the P_{van} promoter. We now state in the strain construction subsection within Material and Methods: “pVMCS6 transformants were selected on the basis of exhibiting constitutive rather than vanillate-inducible GFP signal, indicating that expression was driven by the native promoter.”

In the revised strain list, we have added a new column that helps to describe this study's strain manipulations through the genomic integration of plasmid vectors. This method for strain modification is now clearly separated from the introduction of genes on multicopy plasmids.

Beyond observing that our “normal” strain background, which was actually $\Delta cpdR$ bearing a plasmid containing wildtype *cpdR* under P_{cpdR} plus a genomically integrated copy of ClpX- or RcdA-GFP, appeared phenotypically normal in terms of cell length, stalk formation and polar localization of fluorescently tagged proteins, we did not pursue any in-depth characterization of protein function. This is because the strains were not built for the purpose of showing a wildtype localization profile for ClpX-GFP and RcdA-GFP, but for making relative comparisons of GFP localization profiles under conditions that are abnormal in terms of polar activity (namely, CckA H⁺, $\Delta rcdA$ and CpdRD51A).

For all *C. crescentus* strains expressing mChy-PopZ, the endogenous copy of *popZ* has been replaced by mChy-PopZ, and this coding sequence is under the native *popZ* promoter. This is a functional replacement that rescues the stalkless, filamentous, chromosome segregation and polar localization $\Delta popZ$ phenotypes (Bowman et al. 2012). We added a line in the strain construction section explaining this: “mChy-PopZ complements the severe $\Delta popZ$ mutant phenotypes in the *popZ::mChy-popZ* strain background⁷.”

7. In Figure 4b, it is not clear what the blue and orange histograms represent. Are they the polar localization of stalked versus swarmer? Or are they the fraction of cells with polar localization at the time points shown (in which case, why is it a smooth distribution over time?) . More clarity is needed for this particular subfigure.

The data is intended to present the second of the two possibilities raised by the reviewer's question. To present the histogram information in Figure 5b (formerly 4b) in a more traditional and clear form, we have changed the appearance of the plots from a smoothed line to a bar chart.

The revised figure legend for Figure 5b now states, "The fraction of cells exhibiting a polar focus, normalized to the highest value observed, were plotted on the same time axis (bar charts)." The key at the right side of the panel explains that orange data corresponds to swarmer cells while blue data corresponds to stalked cells.

The error bars in the line graphs show the standard deviation in total fluorescence from 20 different cells (that is, they show the standard deviation of values within the cell population, not the standard deviation of the average value between different trials). The corresponding bar charts, which show the fractions of those cells exhibiting polar foci, do not have error bars because the number is drawn from that dataset. To add error bars to the bar charts, we would have to count additional cells that are not included in the data used to make the line graphs.

8. In Figure 4, the in vivo results results showing rapid degradation of PdeA, TacA and CtrA constructs is very strong and the authors should be complemented for the clarity of the results. It is equally striking that there is still substantial degradation seen in $\Delta popZ$ strains based on the quantified data. Did the authors look at this degradation by microscopy as well? Based on their model, you might imagine there to be no cell division dependence to this turnover.

We use microscopy to track the degradation of CtrA-YFP RD+15 in $\Delta popZ$ cells in Supplementary Figure S5a (a new panel in the revised manuscript) and S5b. $\Delta popZ$ cells divide abnormally and less frequently than wildtype cells. As the reviewer suggests, we find that CtrA-YFP could be degraded outside of any obvious cell division (top right cell in Supplementary Figure 5b). But since degradation also occurred while cells divided, we could measure degradation in dividing cells with the same approach as we used for wildtype cells in main Figure 5. This data is presented in the chart in new Supplementary Figure 5a, which shows that, consistent with the degradation rates in liquid cultures presented in main Figure 5c, the average rate of degradation without *popZ* was slower than we observed in either of the daughter cells types in wildtype cultures (main Figure 5b).

9. In lines 180-185 the authors raise an important concern with using translation inhibitors to evaluate protein degradation. The possibility that chlor addition affects cell cycle progression (and therefore protein stability) is noteworthy. It would be excellent to complement this observation with microscopy experiments showing this same halting of degradation in single cells as well.

We present microscopy data tracking the degradation of CtrA-YFP RD + 15 in wildtype and $\Delta popZ$ cells treated with chloramphenicol in Supplementary Figure 5b, and present quantitative analysis of this information in a new figure panel below the set of images. Since we could observe degradation after chloramphenicol addition, we could measure degradation rates in individual cells, using the same approach as we used in main Figure 5b and Supplementary Figure 5a. Here, the subpopulation of cells that do not rapidly degrade CtrA-YFP (and which never show polar foci of CtrA-YFP) are clearly distinguishable from those that degrade CtrA-YFP as a faster rate (most of which do exhibit polar foci of CtrA-YFP). These observations are consistent with the idea that chloramphenicol has the effect of blocking cell cycle advancement, locking some wildtype cells into a stage that does not rapidly degrade ClpXP substrates.

10. The authors should note the confounding factor of the xylose-based reporter system they are using with respect to cell-cycle regulated expression (such as that for CtrA and TacA). Degradation is controlled during the cell-cycle such that the most robust activity often correlates with the lowest expression of the targets (such as CtrA and TacA). By using a non cell-cycle dependent promoter, the authors may be seeing effects on levels of proteins that is not seen with the physiologically related proteins. Looking at endogenous levels of CtrA and TacA (for example) would resolve this - however, it is not critical to address this experimentally, but it should be commented on in the results.

In wildtype *C. crescentus*, a large fraction of ClpXP substrate degradation occurs during a cell cycle-dependent burst at the swarmer to stalk transition. The data presented in this manuscript and in others (e.g. Ryan Shapiro 2004) support this statement by showing that the cell cycle-dependent pattern of degradation persists when substrate proteins are expressed from a non-cell cycle-regulated promoter. Thus, we conclude that the degradation rates that we are observing are indicative of normal cell cycle-related phenomena. Given that our method of YFP-tagged substrate expression means that substrate proteins are present at higher than normal levels, the reviewer is correct in asserting that the measured rates of substrate degradation may not be fully accurate. However, given that these substrate protein expression conditions are not by themselves sufficient to trigger substantial changes in the cell-cycle dependence of substrate degradation, we feel that the relative comparisons of degradation rates made in this manuscript accurately affect relevant differences across strains and conditions.

11. The simulation results are consistent with the experimentally derived model that dynamic concentration of reactants yields rapid degradation. However, there are a few specifics that should be addressed. There should be more details about the assumptions in the simulation - it is mentioned that some parameters are empirically derived, some experimentally determined. The empirically derived binding radius is 0.025 units, which seems to correspond to 2.5 Å based on the conversions described, but this seems quite small if we are considering the center of particles for these proteins. Is there any biological significance to this value, what is the effect of adjusting this binding radius? Does it affect the diffusion measurements/assumptions? Some explanations here would be excellent.

We agree with the reviewer that the binding radius for the two-component model is quite small and may not be physiologically accurate. As we state in the revised manuscript, "For the simple two-particle model, an empirical process was used to determine a binding radius of 0.025, which yielded a reaction rate that roughly matched the observed rate of protein degradation in *C. crescentus* cells (Fig. 4b)." Note that this reaction is simpler than adapter-mediated degradation in real cells, which requires interactions between more components (adapter, substrate, and ClpXP complexes), and is therefore a more frequent / faster reaction. Thus, we chose an artificially small binding radius in order to account for this difference and obtain a physiological degradation rate.

In a new figure panel (Figure 6c), the revised manuscript now includes simulations of 3 and 4-component substrate degradation reactions. Here, we assigned physiologically realistic on- and off-rates for protein interactions instead of setting interaction radii, and these parameters yielded realistic substrate degradation rates. To achieve these rates in the simulation, Smoldyn translated these parameters to a binding radius of 1.6 nm, which is larger than the binding radius of the two component model, but still small relative to the sizes of the proteins. We provide a possible explanation for this in the revised methods section, which reads, "With these parameters, the binding radius was 1.6 nm, which is less than the radius of the proteins being modeled, meaning that that they will not interact every time they collide. This could reflect conformational and geometric constraints on productive collisions."

12. Along the lines of this simulation, the notion that concentrating reactants to induce faster product formation (in this case degradation) is not altogether surprising. It would be ideal to address the specific use case of the CpdR, RcdA, and PopA dependent degradation that is being experimentally probed. Specifically, can the simulation predict the order of degradation rates for the different substrates that are observed experimentally and referred to in lines 170-174?

As explained in our response to reviewer comment #11, the revised manuscript now includes simulations of 3 and 4-component substrate degradation reactions in a new figure panel (Figure 6c). As described in the associated diagram, the 3-component model simulates a substrate like PdeA, that requires only ClpXP and CpdR adaptor. The 4-component model simulates a substrate like TacA, that requires ClpXP together with CpdR and RcdA as adaptors. Both 3- and 4-component degradation is occurring simultaneously in same cell, using physiological numbers of substrates, ClpXP, and adaptor proteins. This model accurately predicts the order of substrate degradation, with 3-component (PdeA) degradation reaction occurring substantially faster than 4-component (TacA) degradation.

line 167: typo " In stalked cell progeny, all three proteolysis substrates were cleared from the cell between before cell separation"

We corrected this line to read, "In stalked cell progeny, all three proteolysis substrates were cleared in the minutes before physical cell separation"

Reviewer #2 (Remarks to the Author):

In the study titled "Phospho-signaling couples polar asymmetry and proteolysis within a membraneless microdomain in *C. crescentus*", the authors explore the relationship between cell-cycle regulated proteolysis and the polar PopZ condensate in *Caulobacter crescentus*. Three pivotal observations are drawn from this investigation. The main findings of this work are that (i) the phosphorylation state of the protein CpdR, a cell cycle-regulated adaptor protein, affects its localization with the PopZ microdomain, (ii) CpdR further recruits and concentrates the associated members of the ClpXP proteolysis complex into the PopZ microdomain at the poles for timely degradation, (iii) discussion on how localization to PopZ regulates rates of degradation.

Points (i) and (ii) echo some of the previous findings in the field. The authors should consider executing additional experiments to justify their new findings relating to these two points, as listed below. Moreover, critical information that would allow us to assess the work relating to point (iii) is missing.

Below are the points that should be addressed:

Comments related to Figure 1:

1. Lack of evidence for a direct relationship between CpdR phosphorylation state and PopZ: The title of Figure 1, "CpdR phosphorylation state influences its co-localization with PopZ," suggests a direct relationship between the phosphorylation state of CpdR and its co-

localization with PopZ. The data presented illustrate that the polar localization of CpdR is cell-cycle regulated (shown before, Iniesta et al., 2006), that the unphosphorylated CpdR localizes to both poles (shown before, Iniesta et al., 2006), and that the hyper-phosphorylated version doesn't localize to poles (new data). This indicates a correlation between polar localization and phosphorylation state but not necessarily causation with PopZ. Please address this point.

We have revised the title of Figure 1 to read: "Correlations between CpdR's phosphorylation state, polar localization, and co-localization with PopZ."

In addition, the definition of "polar localized" is not clear. How much enrichment of fluorescence signal at the pole accounts for polar localization? Finally, a quantitative analysis of multiple cells in Figures 1e and 1g would offer more clarity.

We have added two new figure panels with quantitative information that describes the results of our fluorescence microscopy of live cells. New Figure 1h provides quantitative analysis of the *C. crescentus* strains described in 1d-g. New panel 4d provides quantitative analysis of the *C. crescentus* strain described in 4b-c.

We now state in the revised methods section that "For image quantitation, a cell was counted as having a polar focus if the appropriate fluorescence channel, when overlaid over the corresponding phase contrast image, displayed a clearly distinguishable increase in fluorescence intensity in the group of pixels in the area of the cell pole relative to the adjacent region of the cytoplasm." We would argue that this is the most common approach toward defining polar localization in *C. crescentus* literature. Two significant issues that arise from defining polar localization in the manner are that i: quantifying polar localization is not always an easy judgement call, and ii: "polar localization" can be quite weak.

Regarding the difficulty of quantifying polar localization, this issue compelled us to hold back from quantifying extremely faint and inconsistently observable foci of RcdA-GFP in the $\Delta cpdR$ strain background. The revised results section states "In a $\Delta cpdR$ background, RcdA-GFP was almost completely diffuse, although many cells retained faint polar foci that were difficult to discern and quantify (Supplementary Fig. 4a)." We provide a magnified image of these faint foci in new supplementary data Figure 4a.

Regarding weak polar localization, we note that this manuscript is one of the first to address the question of whether such weak polar localization may be physiologically relevant. We do this in Figure 6, where we present computational models for substrate protein degradation. The revised results text states: "With these parameters, 14.4% of the total pool of CpdR was concentrated in polar microdomains (Supplementary Fig. 8c), which approximates our *in vivo* observations (17.1 +/- 4.1% of CpdR-YFP was concentrated in polar foci at the 15 minute time point in Fig. 1a)." This section shows how such low levels of polar localization could have a substantial impact on enhancing proteolysis rates. The 4th paragraph of the

discussion section places our findings in relation the broader field's conceptions of "polar localization".

2. Clarification on *cckA* Mutants: From my understanding, the *cckA* mutants (H+ and K-) in this study are expressed using an inducible plasmid while the native *cckA* gene is still present and functional (in strains GB#1980 and GB#1981). This dual perturbation—abundance and kinase activity—should be elucidated in the main text. To better distinguish these effects, perhaps the authors could include a "CckA WT on plasmid" in 1f. I'm assuming "CckA WT" currently denotes a chromosomal copy of *cckA*. This additional strain would use the same plasmid backbone as H+ and K- but express the WT *cckA* gene.

We clarify our method of expressing CckA variants with slightly revised text in the results section: "We asked if we could influence CpdR-YFP phosphorylation and localization by expressing mutant variants of CckA with differing levels of kinase/phosphatase activity from a plasmid²⁹". Further, we have corrected an inconsistency in our labeling in Figure 1. In the original manuscript, we used CckA WT / CpdR-YFP WT to represent strain GB1978, but our labeling is consistent with CckA -- / CpdR-YFP WT, since this strain has no extra copy of CckA. We chose not to include "CckA WT on plasmid", as the reviewer called it, since the WT form of CckA acts more strongly as a kinase than a phosphatase (Chen Laub 2009 PMID: 19783630), and in this experiment, CckA WT on a plasmid produced a localization phenotype that was almost as strong as CckA H+ on a plasmid (not shown). We show the kinase-oriented activity of CckA in our *E. coli* reconstitution experiments in Figure 2.

3. CckA and CpdR Relationship: Given that CckA functions as a kinase for CpdR, the observed increase in phosphorylated CpdR in the H+ mutant, shown in 1f, isn't surprising. However, I find it perplexing that the CckA WT shows a profile close to identical to CckA K-. Moreover, why does there appear to be increased unphosphorylated CpdR_{D51A} in the K- context? A side note: the letter 'K' is absent in the rightmost lane of 1f.

The rightmost lane of 1f has a dash in the row that describes the form of CckA expression in this strain (and, for reasons explained in our response to reviewer #2 comment #2, so does the leftmost lane). We agree that the labels were inconsistent and confusing. To clarify CckA expression in these strains, the revised figure legend now reads: "e, Localization of CpdR-YFP or CpdR_{D51A}-YFP and mChy-PopZ in different CckA signaling contexts. H+ and K- signify hyperactive kinase and kinase deficient forms of CckA expressed from a multicopy plasmid vector in addition to CckA expressed from the unmodified *cckA* locus. Arrowheads mark polar localization. f, CpdR-YFP phosphorylation levels in lysates from e, observed using Phos-tag gel electrophoresis. The dash mark indicates no extrachromosomal copies of *cckA*."

As the reviewer observed, we found that the level of CpdR-YFP phosphorylation in cells with wildtype CckA expression was more similar to cells that expressed K- CckA from a plasmid than cells expressing H+ CckA from a plasmid (Figure 1f). Correspondingly, the localization pattern of CpdR-YFP in cells with wildtype CckA expression more closely resembled cells expressing K- CckA than H+ CckA (new figure 1h). Although the manuscript text does not speculate on why this is the case, we propose that the phosphatase activity of the K- form of CckA is not strong enough to fully overcome the kinase activity of WT CckA produced from the native locus.

The reviewer also noted that the amount of CpdRD51A-YFP was higher in the – strain (no additional CckA) than in the CckA H+ strain (Figure 1f). In the revised manuscript, we now provide three independent trials of this experiment (shown in Figure 1f and new Supplementary Figure 1b). Comparing trials shows that the relative levels of CpdR-YFP in the strains varies between the experiments, while the ratio of phosphorylated to unphosphorylated CpdR (now reported below the gel image in Figure 1f) is fairly consistent. We do not have an explanation for the variability in CpdR-YFP levels. The difficulties raised during attempts to interpret *in vivo* data such as these underscore the need for reductionist approaches that involve fewer proteins and fewer interrelated factors, which we take in Figure 2 and new Figure 3.

4. PopZ Localization in Different Backgrounds: In the H+ context depicted in 1e, PopZ's localization seems confined to a single pole rather than two in dividing *C. crescentus* cells. Is that the case? And if so, how can that be explained? Please add a quantification of PopZ's localization across the varied CckA mutant contexts.

We agree with the reviewer's observation of the originally submitted figure. In the revised manuscript, we have quantified mChy-PopZ and CpdR-YFP localization in this and the other strains shown in figure 1e, and we present the data in new Figure 1h. The data show that about 60% of cells expressing the H+ variant of CckA have bipolar mChy-PopZ foci. This was not accurately represented by the image panels shown in the originally submitted version of 1e. Thus, we have replaced the images of the CckA H+ strain in 1e with more representative data.

5. Phosphogel Replication & Quantification: In relation to the phosphogels presented in Figure 1 (and similarly in Figures 2 and 3), please include three biological replicates. Additionally, please quantify these gel band intensities using standard deviations to improve data robustness.

The revised manuscript now provides three biological replicates of the phos-tag gel experiments shown in 1f and 2b (See new Supplementary Figures 1b and 2a), and two biological replicates of the phos-tag experiment shown in 1b, which provided very similar

results (see new Supplementary Figure 1a). We quantified and averaged the band intensities, and we show the average values and standard deviations for each lane in the associated main figures.

Comments related to Figure 2:

6. PopZ / CpdR Interaction: This work very much relies on the assumption that CpdR directly binds PopZ. While there is data to indirectly support this assumption, the authors do not provide direct evidence. The authors show co-localization imaging assays in *E. coli* and integration of CpdR molecules into PopZ condensates in vitro. As the manuscript's central claim hinges on CpdR's direct binding with PopZ, this must be definitively established in vitro using one of the well-established protein binding assays. It's worth noting that several alternate mechanisms could result in the enrichment of CpdR within PopZ clusters at the cellular pole, especially when the observed enrichment is relatively modest.

We attempted to demonstrate direct interaction between PopZ and CpdR in several different ways, including Surface Plasmon Resonance (weak, inconsistent signal), Isothermal Microcalorimetry (a clear signal, but possibly confounded by additional interactions or multivalency), Microscale Thermophoresis (CpdR crashed out of solution before reaching a high enough concentration for testing), and NMR. The NMR experiments provided clear evidence of a physical interaction between CpdR and PopZ, and we present the data in new Figure 3d and Supplementary Figure 3b and 3c, together with associated new text in the results and methods sections.

7. Condensate images: Concerning the condensate images presented in image 2d, it is difficult to assess the condensate behavior. Please show a comprehensive figure that encompasses multiple condensates for each experimental condition. Alongside this, a detailed quantitative analysis detailing the extent of CpdR's incorporation within these condensates would be beneficial.

In the revised manuscript, we provide an updated image panel of the condensates (new Figure 3a), which shows multiple condensate droplets in the same field.

We also provide more detailed explanations of our quantitative analyses of droplet fluorescence, and we have increased the numbers of droplets that were analyzed (now about 30 in each condition across three separate trials) and also collected data from a wider range of droplet sizes. The revised figure legend now states:

“Average fluorescence intensities of condensates in a, normalized to the value obtained for CpdR-GFP, imaged in YFP channel. 5>n>10 per replicate; bar = standard deviation from 3 replicates; brackets indicate P-values from two-sided unpaired t-test: ** = 0.005, ns=non-significant.”

And the revised methods section states:

“Condensates of varying size from were analyzed using Zeiss Zen Blue software. The spline tool was used to draw circles around the condensates, and fluorescence values were quantified by dividing the total pixel intensity by the two dimensional area.”

8. The authors suggest in line 122: "In conjunction with the *E. coli* reconstitution experiments, these findings imply that CpdR phosphorylation may hinder its direct interaction with PopZ." While this is an intriguing interpretation of the presented data, it remains speculative without additional experimental corroboration.

The revised manuscript now includes a conventional method for demonstrating a direct interaction between CpdR and PopZ: chemical shift perturbation analysis in NMR (Figure 3d and Supplementary Figure 3b and 3c). It also includes another *in vitro* protein association assay, based on CpdR-GFP partitioning into phase-separated PopZ condensates, which shows that CpdR phosphorylation reduces its interaction with PopZ (Figure 3a-c., Supplementary Figure 3a). The manuscript also shows that the presence of an active CpdR phosphorylation mechanism inhibits CpdR's interaction with PopZ condensates in *E. coli* cytoplasm (Figure 2a-c, Supplementary Figure 2a-c.), and we use two different *in vivo* methods to show that CpdR's phosphorylation state in *C. crescentus* is correlated with its polar localization and co-localization with polar PopZ foci (Figure 1).

The revised manuscript includes a modified version of the sentence in question, “These results suggest that CpdR phosphorylation inhibits its physical interaction with PopZ.” We believe that the evidence presented in the revised manuscript supports this statement.

9. Interpretation of Figure 2b, like many of the other imaging data, would benefit from quantitative analysis of many cells. Certain instances display a CpdR focus without a concurrent PopZ focus, as observed in the top panel with WT CckA and CckA K-. This leads to the following queries: (i) Can CpdR foci be discerned even when PopZ is not prominently visible or does the mere presence of a minimal PopZ focus suffice to localize CpdR? (ii) Is it possible for CckA to play a role in CpdR polar localization that is independent of PopZ?

The revised figure legend indicates the number of cell analyzed, “Normalized fluorescence intensities were plotted against cell length (n=60, with 20 cells from 3 replicates).” We found that adding additional cells or experimental replicates had very little effect on the charts' appearance.

We agree that the originally submitted image panels were not ideal. Brightness/contrast adjustment issues gave the mistaken impression that CpdR-GFP foci could occur at locations where there no mChy-foci. In fact, we do not observe any CpdR-GFP foci in *E. coli*

cells in the absence of mChy-PopZ, even when CckA and ChpT are also present (revised Main Fig. 2c and new Supplementary Fig. 2c). We present new image panels in the main figure image panels, with improved brightness-contrast adjustment, to avoid giving the impression of CpdR-GFP foci where there is no mChy-PopZ.

The revised manuscript also includes an additional control experiment for the *E. coli* co-expression assay, shown in new Supplementary Figure 2b. Here, we show that the co-expression of CpdR with ChpT, another direct binding partner of PopZ, does not affect the PopZ-CpdR interaction in the absence of CckA kinase and the associated CckA-ChpT-CpdR phosphorelay.

Comments relating to Figure 3

10. Please add quantification across many cells for the co-localization data shown in Figures 3c and 3d. It is very difficult to interpret the data as presented.

New Figure 4d provides quantitative analysis of the *C. crescentus* strains shown in Fig. 4c (formerly Fig 3c-d). We provide information on the number of experimental replicates and the number of cells quantified per replicate in the updated figure legend: "d. The average percentage of cells with diffuse, monopolar, and bipolar fluorescent foci in c (n>100/replicate, bar = standard deviation of 3 replicates)."

Due to the faintness of the fluorescent foci, we were unable to quantify one of the strains. Please refer to our response to Reviewer #2, Comment #1 for our commentary on the brightness / weakness of polar foci in this particular strain and, in a broader sense, the meaning of "pole localized" for proteins in *C. crescentus*.

11. Some of the data presented as part of this figure was already shown, including that CpdR controls ClpXP localization to the pole (Iniesta et al. 2006) and that CpdRD51A leads to bi-polar localization of ClpXP (Iniesta et al. 2006). The new data here is that hyper-phosphorylated alters RcdA and ClpXP localization. However, more quantitation is required to support these claims, as indicated in point (10).

Quantitation of fluorescence microscopy images was addressed in our response to the previous comment.

We have revised the results text to more prominently credit the Iniesta 2006 study that observed CpdR's influence on ClpX localization. We note that the revised manuscript not only agrees with but also bolsters that study, by providing quantitative analysis on ClpX-GFP polar foci in a $\Delta cpdR$ background and where ClpX-GFP is expressed from its native promoter, quantitative analysis of polar foci for the whole cell population (not just

swarmers) in CpdRD51A-GFP expressing strains, and correlative information on the polar localization pattern of mChy-PopZ in these strains.

In response to reviewer comments, we made minor revisions to the paragraphs describing our fluorescence microscopy data. We quote those line below, with revisions indicated by blue colored text:

“We hypothesized that we could control the localization of ClpXP complexes (Fig. 4a) by modifying the phosphorylation state of CpdR. To test this, we created strains in which either the ClpX-associated adaptor RcdA¹⁸ or ClpX itself were tagged with GFP and expressed from their endogenous promoters, and in which the native copy of *popZ* had been exchanged with a functional mChy-tagged version⁷ (Fig. 4b). In an otherwise wildtype genetic background, RcdA-GFP and ClpX-GFP both exhibited the expected pattern of transient co-localization with polar mChy-PopZ during the swarmer-to-slaked cell transition^{18,30} (Fig. 4c-d). In a $\Delta cpdR$ background, RcdA-GFP was almost completely diffuse, although many cells retained faint polar foci that were difficult to discern and quantify (Supplementary Fig. 4a). In agreement with an earlier report¹⁷, we also observed that CpdR is required for ClpX polar localization (Fig. 3c).

To assess protein localization in cells where CpdR is always de-phosphorylated, we replaced endogenous *cpdR* with the *cpdR_{D15A}* variant. Strikingly, RcdA-GFP was localized to both poles in cells that also had bi-polar mChy-PopZ foci, and, in agreement with an earlier report¹⁷, ClpX-GFP showed a similar, though less intensely localized pattern (Fig. 4c-d). To assess protein localization in cells where CpdR is always phosphorylated, we expressed the hyperactive variant of CckA. In these cells, RcdA-GFP and ClpX-GFP were diffuse. When we expressed hyperactive CckA in the *cpdR_{D51A}* background, RcdA-GFP and ClpX-GFP exhibited bi-polar localization, indicating that CpdR de-phosphorylation is a controlling factor in these proteins' polar localization. The differences in these strains' localization patterns suggest that in normal cells, CpdR's phosphorylation state is under strict temporal and spatial control, directing ClpXP complexes to one pole at the swarmer-to-slaked cell transition. Since ClpX is reported to play a key role in RcdA-GFP polar localization¹⁸, CpdR may promote RcdA localization indirectly, by recruiting ClpX to the pole. Additional RcdA may be localized to poles via direct contact with PopZ¹⁹.

12. In addition, Nordyke et al. 2020 showed by NMR that RcdA directly binds PopZ. Can the authors explain why and how CpdR phosphorylation state seems to affect RcdA recruitment to PopZ?

The reviewer raises an interesting and somewhat perplexing question. Stated slightly differently, How can CpdR be responsible for RcdA's polar localization if i:RcdA binds directly to PopZ (Nordyke et al. 2020), and ii:RcdA is no longer pole localized in a ClpX depletion strain (McGrath et al. 2006)?

The revised manuscript provides a more nuanced view of RcdA-GFP localization in the $\Delta cpdR$ background. As we state in the revised text, “In a $\Delta cpdR$ background, RcdA-GFP was almost completely diffuse, although many cells retained faint polar foci that were difficult to discern and quantify (Supplementary Fig. 4a).” The presence of faint RcdA-GFP polar foci in the $\Delta cpdR$ strain indicates that CpdR is not the only factor that directs RcdA-GFP to cell poles. In the next revised paragraph of the results section, we propose that CpdR promotes “robust” RcdA localization indirectly via ClpXP localization, leaving room for an additional factor. Based on the findings

of the Nordyke et al 2020 study, we hypothesize that the other factor is a weak interaction between RcdA and PopZ.

Comments relating to Figure 4

13. In Fig. 4b, the authors have shown that the polar localization of three proteolysis substrates, PdeA, TacA, and CtrA-RD+15, governs their time of degradation in the swarmer and stalked cell progeny. Please explain why the polar localization of the substrates differs between the swarmer and stalked cells. What are the factors that contribute to that? Also, what exactly drives the proteolysis in the swarmer cell where CpdR does not localize in the PopZ microdomain?

C. crescentus cells divide asymmetrically into swarmer cells and stalked cells. The swarmer cells inherit a suite of regulatory factors (many pole-localized) that communicate a delay in cell cycle progression. By contrast, stalked cells inherit a suite of regulatory factors (many pole-localized) that communicate an immediate entry into S-phase and associated cell cycle advancements. These polar factors include an oppositely localized histidine kinase / phosphatase pair that controls a regulator of CckA (Childers et al. PMID: 25349992, Paul et al PMID: 18455986), oppositely localized upstream factors that regulate those kinases and other pathways (Saurabh et al. 35171683, Chong et al. PMID: 38055339, Zhang et al. PMID: 35124010), and a pole-localized diguanylate cyclase that may also contribute to CckA control via c-di-GMP signaling (Lori et al. PMID: 25945741). Since the upstream signaling that drives cell cycle advancement and associated ClpXP-mediated proteolysis is quite complex, we decided not to describe it in detail in the manuscript text. Instead, the revised manuscript text directs readers to helpful references on this subject.

Quoting from the revised manuscript text and highlighting revisions in blue colored text: "In stalked cell progeny, which are programmed to advance immediately to S-phase and subsequent stages of the cell cycle³⁴⁻³⁶, all three proteolysis substrates were cleared in the minutes before physical cell separation, and the highest frequency of polar localization was observed during this time period. Since cell separation lags the separation of progeny cells' cytoplasm via inner membrane fusion by several minutes³⁴, it is likely that proteolysis and polar localization occurred soon after compartmentalization. In swarmer cell progeny, which are developmentally delayed relative to stalked cell siblings, YFP-PdeA was often cleared concomitantly or within four minutes of its clearance from stalked cells, which coincided with the highest frequency of polar localization. The majority of YFP-TacA and YFP-CtrA RD+15 were cleared approximately 12 minutes later, when their peak frequency in polar localization occurred (Fig. 5b)..."

Regarding the reviewer's question of what drives proteolysis in cells where there is no obvious polar focus, we point out that ClpXP can be active whether it is localized at a cell pole or not. As we show in Supplementary Figure 5, substrate proteolysis still occurs in the absence of PopZ. In this manuscript, our goal is to show that the rates of substrate protein degradation by adaptor-mediated ClpXP proteolysis is substantially enhanced (but not

turned on in the sense of a switch) when the components are concentrated within PopZ microdomains. Although we could not find a convincing way of demonstrating this in the manuscript, we also point out that nearly all cells show polar foci of CpdR-YFP and substrates at some point during the time of degradation in time lapse movies, but the localization can be faint, transient, and not present in some time frames, often depending on image focal plane and the time interval between exposures.

14. Please indicate that the three substrates (PdeA, TacA, and CtrA-RD+15) are all expressed on inducible plasmids. This is not clear in the main text and is confusing to the reader.

Coding sequences for YFP-PdeA, YFP-TacA, YFP-CtrA-RD+15, HA-PdeA, HA-TacA, and HA-CtrA-RD+15 were inserted into the *C. crescentus* chromosome by single-copy genomic integration, and expressed under control of a xylose-inducible promoter. We modified *C. crescentus* strains with the genomically integrating vector pXMCS-5, where the P_{xyl} promoter sequence is functional but only 156 base pairs in length, and therefore too short to be used as a site for genomic integration at the *xyl* locus. Consequently, this greatly favors integration at the *pdeA*, *tacA*, or *ctrA* loci. Since the YFP or HA tag is at the N-terminal end of the coding sequence, genomic integration results in a merodiploid locus that includes one native untagged copy of the gene and one additional YFP or HA-tagged coding sequence under control of the mini P_{xyl} promoter.

We now specify the use of genomically integrating vectors in the revised strain list. We also state in the strain construction subsection within Material and Methods: "pXMCS5 transformants were selected on the basis of xylose-inducibility."

15. The discussion in lines 180-185 is unclear. The authors write, "... we blocked new protein synthesis by treating cells with chloramphenicol. Over subsequent time points, a subpopulation of cells in wildtype cultures retained large quantiles of substrates for more than 60 minutes, but this was not observed in $\Delta popZ$ cultures (Extended Data Fig. 3a)." That means that protein degradation works better in $\Delta popZ$ cells in this experiment. They then write: "We propose that chloramphenicol treatment interferes with the measurement of degradation rates by preventing cell cycle progression in swarmer cells, locking them into a stage of high substrate stability. $\Delta popZ$ strains, whose cell division is often uncoupled from the cell cycle, do not appear to produce significant numbers of this cell type." There is no evidence in this paper to substantiate these claims, and other mechanisms are likely possible. This data is confusing, and the authors should consider either removing it or providing additional experiments to explain these results.

We feel that it is important to keep the data on the difference in substrate protein degradation kinetics in normal versus chloramphenicol-treated cells because they agree

with a previous study (cited in the manuscript) that shows that $\Delta popZ$ cells degrade substrates more quickly after chloramphenicol treatment. This is important because our goal is to compare and contrast studies of substrate proteolysis in $\Delta popZ$ cells in published literature with the results shown in this manuscript, which reveals a different result when substrate proteolysis is measured without chloramphenicol.

In this manuscript, we make the point that the observations that we have made without chloramphenicol treatment are more relevant to normal cell physiology, since these conditions allow cell division and cell cycle progression. Additionally, we feel that it is important to explain how chloramphenicol treatment, at least when the whole cell population is considered, produces conflicting results.

To more accurately demonstrate the reduction in substrate proteolysis rates in wildtype cells after chloramphenicol treatment, we have added new quantitative analysis to the revised manuscript (new Supplementary Figure Panel 5c). The data clearly shows that "a subpopulation of cells in wildtype cultures retained large quantiles of substrates for more than 60 minutes, but this was not observed in $\Delta popZ$ cultures (Supplementary Fig. 5b-c)", as stated in the results section of the manuscript.

We agree that our proposed explanation for the observed difference between $\Delta popZ$ and wildtype cells after chloramphenicol treatment is speculative. We note, however, that the manuscript provides some support for our claim that the governance of cell cycle progression is relatively weak in $\Delta popZ$ cells in new Supplementary figure S5a, which shows that the difference in the rate of CtrA-YFP degradation in the two daughter cells in $popZ$ cells is small compared to wildtype cells, shown in main Figure 5b. Despite this problem, we feel that including the chloramphenicol experiments is preferable to omitting them. By omitting them, the manuscript would merely pose a rebuttal of the previous study without attempting to show why the new study provides more physiologically relevant information.

16. In lines 187-188, the authors write, "Under these conditions, ClpXP substrates were degraded up to twice the rate in wildtype cells compared to delta $popZ$." Looking at Figure 4c, this does not seem to be the case. TacA and PdeA show modest to no effect in clearance in WT vs. delta $popZ$ cells, at least in the blots presented. For the case of CtrA, we know that its degradation is phosphorylation-state dependent. Can the disruption of CtrA dephosphorylation affect the slower rate of its degradation? Also, the authors say that the quantitation is based on three biological replicates. Please show these replicates in SI.

Replicates of the HA-tagged substrate degradation experiments are shown in new Supplementary Figure 7. In response to this reviewer comment, we have modified our description of the results of these experiments. We now state, "Under these conditions, the three

ClpXP substrates that we tested were degraded more rapidly in wildtype cells compared to $\Delta popZ$. The difference was higher for substrates whose degradation is mediated by more adapter proteins: CtrA requires three adapters (CpdR, RcdA, and PopA), TacA requires two (CpdR and RcdA), and PdeA requires only CpdR²³." Notably, these results are consistent with our computational model, which shows that the degradation rate of a substrate with two PopZ-associated adaptors is strongly enhanced by the presence of a PopZ microdomain, while a substrate that interacts with only one PopZ-associated adaptor is not strongly affected by the absence of PopZ.

Regarding the effect of CtrA phosphorylation, the literature shows that the active phosphomimetic form of CtrA and the dephosphorylated form are both efficiently degraded in *C. crescentus* (Ryan and Shipiro 2002 PMID: 12445780, Domian et al. 1997 PMID: 9267022), indicating that CtrA's phosphorylation state is not driving the difference in its rate of degradation.

Comments relating to Figure 5

17. The authors have used simulations to show how increasing the concentration of substrates within the PopZ microdomain enhances the rate of proteolysis. Many details of binding rates are missing, to the point that it is difficult to assess the quality of this section.

In the revised manuscript, we now present two computational models. The first model is the same as the one presented in the original manuscript submission. Here, the binding rates are defined by setting a "reaction radius" for the yellow and red particles. Reaction radii in Smoldyn simulations have no formal relationship to biophysical rate constants or binding affinities, particularly for a non-reversible reaction like the one in the first model.

However, for reversible reactions, when Smoldyn is supplied with diffusion rates and time steps, the program can translate reaction radii to standard kinetic rate constants. The second Smoldyn model that we present in the revised manuscript includes reversible reactions between particle species representing CpdR, ClpXP, RcdA, and substrates (new Figure 6c). We assigned physiological rate constants that correspond to a dynamic, low affinity interaction, and Smoldyn translated these parameters into a reaction radius for the simulation.

As we describe in the updated Methods Section: "For the two-particle models in Figure 6a-b, an empirical process was used to determine a binding radius of 0.025, which yielded a reaction rate that roughly matched the observed rate of protein degradation in *C. crescentus* cells (Fig. 5b). For the complex model in Figure 6c, Smoldyn calculated the binding radii from user-defined values for on-rate and off-rates. Since the binding affinities are unknown, we used the same values for all particle interactions ($1 \times 10^4 \text{ M}^{-1} \text{ sec}^{-1}$ / $0.166 \text{ px}^3 \text{ msec}^{-1}$ for on-rate and 10 sec^{-1} / 0.01 msec^{-1} for off-rate), which are well within the range

of measured rate constants for dimeric complexes⁵³ and correspond to a weak association with a K_D of 100 micromolar.”

As described in the Results section, interactions with PopZ were simulated by reducing the diffusion rate of particles in areas that were defined as polar microdomains, using information from single molecule tracking studies that measured diffusion coefficients for proteins when diffusing in the main cytoplasmic compartment and when located in polar microdomains. The revised results section states: “With these parameters, 14.4% of the total pool of CpdR was concentrated in polar microdomains (Supplementary Fig. 8c), which approximates our *in vivo* observations (17.1 +/- 4.1% of CpdR-YFP was concentrated in polar foci at the 15 minute time point in Fig. 1a).”

18. In extended data controls, I would also like to see that the simulations recapitulate the data presented in Figures 1-3. For example, CpdR, ClpXP, and RcdA localization in the different mutational backgrounds.

In the revised manuscript, we present a second, more complex computational simulation, where the diffusing particles are given properties that make them resemble CpdR, ClpXP, RcdA, and the substrates PdeA and TacA (new Figure 6c and Supplementary Figure 8c).

The primary use of the model was to test the influence of PopZ microdomains on substrate proteolysis rates, and this is described in the main text and main Figure 6c. We could also observe the localizations of CpdR, ClpXP, and RcdA in different “mutant” backgrounds, and these data are shown in Supplementary Figure 8c. We investigated the consequences of eliminating the PopZ-CpdR interaction (an aspect of the CckA H+ condition, termed CpdR^{mut} in the figure) by simply “turning off” the reduction in diffusion rate of CpdR and CpdR-associated particles in polar microdomains. This eliminated the concentration of CpdR and ClpXP at the cells poles, and reduced the proteolysis rate of TacA substrate.

Eliminating CpdR simulated the conditions of a $\Delta cpdR$ strain. This had the effect of inhibiting proteolysis and de-localizing ClpXP (as we observed in Figure 3), but unlike our observations in *C. crescentus*, RcdA localization was unaffected. Clearly, this simplified model is lacking some features that are present in the natural system. It could be that RcdA’s interaction with PopZ is relatively weak, and that more of its polar localization comes from its association with ClpXP rather than with PopZ (see also our comment on RcdA localization in our response to Reviewer #2 comment #12). We note, however, that the main purpose of these models is to provide support for the idea that the concentration of substrates, adaptor proteins, and ClpXP in polar microdomains could enhance proteolysis rates in *C. crescentus*. We do not intend to present a complete model or demonstrate that the process is fully understood.

Other comments:

19. In Fig. 1b, the authors should provide error bars as it will give a better idea of the variation in the data.

The revised figure panel includes error bars.

20. In Fig. 2e, the authors should replace the gel image with a cleaner version to better understand the bands.

Figure 2e is moved to Figure 3b in the revised manuscript. We apologize for the uneven appearance of the gel, which comes from the reduced performance of Phos-tag gels compared to standard PAGE, particularly in resolving cell lysates. The revised manuscript now includes gels from three independent trials, two of which are presented in Supplementary Figure 3a.

21. The majority of figures in the extended data figures lack scale bars.

Scale bars have been included in the image panels of the revised figures.

22. Significant proofreading of the manuscript is required.

We apologize for the typos. We have proofread the text and corrected a few.

Reviewer #3 (Remarks to the Author):

In this manuscript, the authors characterise the dynamic localisation of polar proteins in the bacterial model *Caulobacter crescentus*. The authors were interested in understanding how proteins, such as the response regulator CpdR, are localised at one pole while interacting with the bipolar hub PopZ. In particular, they show data strongly suggesting that only unphosphorylated CpdR interacts with PopZ. Based on that, they propose that the PopZ:CpdR interaction is driven by the asymmetric activity of the the bipolarly localised histidine kinase CckA. Indeed, CckA was shown to work as a kinase at the "new" pole and as a phosphatase at the "old" pole, thereby increasing the local concentration of unphosphorylated CpdR at the old pole, that is where it interacts with PopZ. On the other hand, they show that this PopZ-dependent unipolar localisation of CpdR enhances rapid ClpXP-dependent proteolytic degradation of substrates at the old pole. The question raised in this work is original and important. Although the data support the conclusions, then manuscript could be improved (see below).

Major comments

- First, I don't understand why the authors only quantified the localisation of CpdR-YFP (Figure 1b). I would suggest to quantify all the localisation data (Fig 1e, 1g, 2b, 3c, 3d) instead of only showing a few cells. This is particularly critical when differences are tiny. For instance, the authors state "When we expressed a kinase-deficient (K-) form of CckA (H322A), the ratio was substantially lower, and cells exhibited robust polar foci." Terms like substantially or robust do not indicate to what extent the effect is observed.

We have added two new figure panels with quantitative information that describes the results of our fluorescence microscopy of live cells. New Figure 1h provides quantitative analysis of the *C. crescentus* strains described in 1d-g. New panel 4d provides quantitative analysis of the *C. crescentus* strain described in 4b-c. For these and other panels describing quantitative analysis of cell images (1b, 2c, S2b), we also provide information on the number of experimental replicates and the number of cells quantified per replicate in the associated figure legend.

Due to the faintness of the fluorescent foci, we were unable to quantify one of the strains. Please refer to our response to Reviewer #2, Comment #1 for our commentary on the brightness / weakness of polar foci in this particular strain and, in a broader sense, the meaning of "pole localized" for proteins in *C. crescentus*.

All the phos-tag gels should also be quantified (Fig 1c, 1f, 2c) to confirm the effects expected from the mutants used. For example, I don't see a substantial change in the phosphorylated/unphosphorylated ratio of CpdR in the K- mutant of CckA on figure 1f.

The revised manuscript now provides three biological replicates of the phos-tag gel experiments shown in 1f and 2b (See new Supplementary Figures 1b and 2a), and two biological replicates of the phos-tag experiment shown in 1b, which provided very similar results (see new Supplementary Figure 1a). We quantified and averaged the band intensities, and we show the average values and standard deviations for each lane in the associated main figures.

The ratio of phosphorylated to unphosphorylated CpdR-YFP in cells expressing the CckA K- mutant (in addition to endogenous CckA) was 0.47, compared to 0.55 in cells that express endogenous CckA only (Figure 1f). This slight difference corresponds with a slight difference in the polar localization of CpdR-YFP (Figure 1h). Cells expressing CckA K+ and CpdRD51A mutants have greater differences in the phosphorylation ratio, and correspondingly show greater differences in the polar localization of CpdR-YFP.

Refer to reviewer #2 comment #2 for more detail on how CckA was expressed in our *C. crescentus* strains.

- I would also suggest the authors to strengthen their evidence of a direct interaction between unphosphorylated CpdR and PopZ. First, they should quantify the fluorescence data shown in figure 2d.

In the revised manuscript, we provide an updated image panel of the condensates (new Figure 3a), which shows multiple condensate droplets in the same field.

We also provide a more detailed explanation of our quantitative analyses of droplet fluorescence in the revised figure legend. We have increased the numbers of droplets that we analyzed (now about 30 in each condition across three separate trials) and taken data from a wider range of droplet sizes. The figure legend of the revised manuscript now states:

“Average fluorescence intensities of condensates in a, normalized to the value obtained for CpdR-GFP, imaged in YFP channel. $5 < n < 10$ per replicate; bar = standard deviation from 3 replicates; brackets indicate P-values from two-sided unpaired t-test: ** = 0.005, ns=non-significant.”

The methods section provides additional information on this analysis: “Condensates of varying size from were analyzed using Zeiss Zen Blue software. The spline tool was used to draw circles around the condensates, and fluorescence values were quantified by dividing the total pixel intensity by the two dimensional area.

Then, they could purify native CpdR (WT or point mutants), instead of CpdR-YFP derivatives, and perform in vitro assays which allows measurements of affinity constants between purified PopZ and CpdR variants. For example, ITC was already used to demonstrate direct interaction between PopZ and ZitP (Bergé et al., 2016 DOI: 10.7554/eLife.20640).

We attempted to demonstrate direct interaction between PopZ and CpdR in several different ways, including Surface Plasmon Resonance (weak, inconsistent signal), Isothermal Microcalorimetry (a clear signal, but possibly confounded by additional interactions or multivalency), Microscale Thermophoresis (CpdR crashed out of solution before reaching a high enough concentration for testing), and NMR. The NMR experiments provided clear evidence of a physical interaction between CpdR and PopZ, and we present the data in new Figure 3d and Supplementary Figure 3b and 3c, together with associated new text in the results and methods sections.

Maybe a phosphomimetic mutant of CpdR (D51E) could also be constructed and characterised? Such a mutant should display opposite phenotypes and behaviours to the D51A mutant.

We constructed the D51E variant of CpdR and tested its interaction with PopZ in the *E. coli* co-expression assay, expecting that, if it is an effective phosphomimetic, it should not to co-localize with PopZ in polar foci. Unfortunately, this was not the case, suggesting that this variant of CpdR does not behave as a phosphorylated protein. This is consistent with the lack of any reports on the CpdR51E variants in the literature, and with the fact that phosphomimetic substitutions do not always give a hyperactivating phenotype (PMID: 14563873, PMID: 14731287).

- The authors suggest that CpdR may influence RcdA localisation indirectly, via ClpX. To check that, RcdA-GFP could be localised in a cpdRD51A Δ socAB Δ clpX. Alternatively, RcdA_ Δ C, which cannot interact with ClpX::CpdR (Joshi et al., 2015 <http://dx.doi.org/10.1016/j.cell.2015.09.030>)

We constructed a Δ socB Δ clpX RcdA-GFP strain and observed that approximately 50% of (unsynchronized) cells exhibited polar foci of RcdA-GFP. This was a surprise, given the observation that RcdA-GFP is diffusely localized after 6 hrs of ClpX depletion in an otherwise wildtype background (Ref #18).

Since this manuscript doesn't focus on the mechanisms for RcdA localization, we decided not to pursue this discrepancy. We have removed our proposal for the order of recruitment of polar factors from the revised manuscript. We deleted the following sentence:

"Since RcdA-GFP requires ClpX for polar localization¹⁸, CpdR may influence its localization indirectly. We propose the following order of protein recruitment at *C. crescentus* cell poles: PopZ -> dephosphorylated CpdR -> ClpX -> RcdA."

..and replaced it with:

"Since ClpX is reported to play a key role in RcdA-GFP polar localization¹⁸, CpdR may promote RcdA localization indirectly, by recruiting ClpX to the pole. Additional RcdA may be localized to poles via direct contact with PopZ¹⁹ "

Minor comments

- The authors should explain how the strains were constructed and designed. For example, how could a CckA kinase-deficient strain survive knowing that cckA is essential? Some important details are missing

We clarified the results text by mentioning that CckA is expressed from a plasmid in the strains shown in Figure 1. The new text reads: "We asked if we could influence CpdR-YFP phosphorylation and localization by expressing mutant variants of CckA with differing levels of kinase/phosphatase activity from a plasmid²⁹."

We have also modified our strain list to clarify strain construction. The table now contains an extra column that allows us to distinguish between strains with replicating plasmids (as in the case of CckA expression) and those that have been modified by genomic integration of non-replicating vectors. Note that the endogenous copy of CckA is left intact in all of our strains, as indicated by the absence of any *cckA* modifications in the Genotype / Background column.

To further clarify CckA expression in these strains, the revised legend of Figure 1 now reads: "e, Localization of CpdR-YFP or CpdR_{D51A}-YFP and mChy-PopZ in different CckA signaling contexts. H+ and K- signify hyperactive kinase and kinase deficient forms of CckA expressed from a multicopy plasmid vector in addition to CckA expressed from the unmodified *cckA* locus. Arrowheads mark polar localization. f, CpdR-YFP phosphorylation levels in lysates from e, observed using Phos-tag gel electrophoresis. The dash mark indicates no extrachromosomal copies of *cckA*."

For additional discussion of our construction of *C. crescentus* strains expressing CckA, refer to our response to Reviewer 2 comment #2.

- Lines 183-185: the interpretation of the results shown in Extended Data Fig. 3a is hard to understand, that is chloramphenicol interfere with degradation rates by preventing cell cycle progression. It looks like rather an hypothesis than an argument. Is there a way to check if this hypothesis is correct?

Despite the fact that our interpretation of the chloramphenicol results are somewhat speculative, we feel that it is important to keep the data on the difference in substrate protein degradation kinetics in normal versus chloramphenicol-treated cells because they agree with a previous study (cited in the manuscript) that shows that $\Delta popZ$ cells degrade substrates more quickly after chloramphenicol treatment. This is important because our goal is to compare and contrast studies of substrate proteolysis in $\Delta popZ$ cells in published literature with the results shown in this manuscript, which reveals a different result when substrate proteolysis is measured without chloramphenicol.

In this manuscript, we make the point that the observations that we have made without chloramphenicol treatment are more relevant to normal cell physiology, since these conditions allow cell division and cell cycle progression. Additionally, we feel that it is important to explain how chloramphenicol treatment, at least when the whole cell population is considered, produces conflicting results.

To more accurately demonstrate the reduction in substrate proteolysis rates in wildtype cells after chloramphenicol treatment, we have added new quantitative analysis to the revised manuscript (new Supplementary Figure Panel 5c). The data clearly shows that “a subpopulation of cells in wildtype cultures retained large quantiles of substrates for more than 60 minutes, but this was not observed in $\Delta popZ$ cultures (Supplementary Fig. 5b-c)”, as stated in the results section of the manuscript.

We agree our reviewers that our proposed explanation for the observed difference between $\Delta popZ$ and wildtype cells after chloramphenicol treatment is speculative. We note, however, that the manuscript provides some support for our claim that the governance of cell cycle progression is relatively weak in $\Delta popZ$ cells in new Supplementary figure S5a, which shows that the difference in the rate of CtrA-YFP degradation in the two daughter cells in $\Delta popZ$ cells is small compared to wildtype cells, shown in main Figure 5b. Despite this problem, we feel that including the chloramphenicol experiments is preferable to omitting them. By omitting them, the manuscript would merely pose a rebuttal of the previous study without attempting to show why the new study provides more physiologically relevant information.

- Line 74: phosphorylated → phosphorylated
- Line 79: unphosphorylated → unphosphorylated

Thank you for pointing out these typos. They have been corrected.

Reviewer #2 (Remarks to the Author):

The authors have effectively addressed the comments raised by the reviewers. However, one major issue must be resolved before the paper can be considered for publication: the in vitro data shown in Figure 3b-d. The authors used partitioning of CpdR-GFP into PopZ condensates to determine whether CpdR interacts with PopZ. This strategy assumes that enrichment of CpdR within the condensate indicates binding. Although valid, the implementation used in this work has issues. The authors measured the enrichment of CpdR mutants by comparing the average total GFP intensity within a PopZ droplet to that of CpdR WT. This metric is not standard practice in the field, which favors partition coefficient measurement (examples include PMID: 33169001, PMID: 33395293, PMID: 35245500).

Response: The newly revised manuscript now shows a partition coefficient, which is the ratio of the concentration of molecules in the condensate to those outside of the condensate (as measured by GFP fluorescence intensity in these two areas).

There are several problems with this method.

First, the distribution of CpdR inside PopZ condensates, as indicated by the six condensates presented, is not straightforward. In two cases, CpdR appears to be uniformly distributed within the condensate, while in four cases, it decorates either the full or partial peripheral ring around the entire droplet. In the cases where the signal is only on the peripheral ring (which would still indicate binding), is the average taken on the whole condensate or only the outer ring?

Response: For reasons described in the new revision of the methods section, we have decided to focus on the intensity of the peripheral rings. We note that, within a given sample, the intensities of the peripheral rings were more consistent between droplets than other fluorescence measurements, such as total droplet intensity. Smaller droplets had relatively higher total fluorescence per area, probably because of the influence of bright out-of-plane peripheral rings that were near the center of the droplet. We have replaced the images in Figure 3b with new images from new repetitions of the experiment, which show more droplets, more detailed ring fluorescence, and the surrounding pool of unbound CpdR-GFP.

The methods section of the newly revised text now reads:

Condensates of varying size were analyzed using Zeiss Zen Blue software. The linescan tool was used to draw a 10 μm line through the middle of each condensate, which was then separated into 160 bins of equal length. The fluorescence intensity of the bright ring of CpdR-GFP at the periphery of the condensate was obtained by taking the median value of the brightest 10 bins. The fluorescence intensity of the inner area of the condensates appeared to be affected by out-of-plane fluorescence, particularly in small condensates, and was therefore omitted from the analyses. The fluorescence intensity of the unbound pool of CpdR-GFP molecules was taken as the median value of the bins from outside the condensate. After subtracting background signal (observed in mounts without CpdR-GFP), we calculated the ratio of the fluorescence intensity of the bright peripheral ring of CpdR-GFP to the fluorescence intensity of the unbound CpdR-GFP molecules in the surrounding medium. To account for sample thickness, which altered fluorescence intensity from the out-of-focus pool above and below the condensates, we used the formula $\frac{(R - (P - A))}{(P - (P - A))}$, Where R is peripheral ring intensity, P is the local unbound pool intensity, and A is the average intensity of the unbound pool across all condensates in the sample set. The value for A differed by less than 22% between samples.

Second, for phosphorylated CpdR (CpdR-GFP+AcP), accumulation is clearly seen on the peripheral ring of the condensate. The signal is less bright, which could be due to lower affinity, different imaging settings, or inherent differences in PopZ condensates in response to AcP. I suspect that using a partition coefficient measurement, which quantifies the fraction of phosphorylated CpdR within condensates, would show low-affinity binding compared to the current approach, which considers relative changes in fluorescence intensity.

Response: The newly revised quantitation method shows a substantially lower partitioning coefficient for CpdR-GFP+AcP compared to un-phosphorylated CpdR-GFP. Note that the image capture and brightness/contrast adjustment settings are the same for all images in this figure panel.

Finally, in this dataset, the external area of the condensate is consistently black, including in the GFP control, which would be expected to show a uniform yellow signal throughout.

Response: Indeed, there is a uniform pool of unbound CpdR-GFP in all conditions, which was not visible in earlier versions of this figure because of image adjustment settings. In the newly revised figure panel, image/contrast is adjusted to show this unbound pool in the fluid surrounding the condensates.

For these reasons, the data presented in Figure 3b-d does not conclusively support the conclusion that CpdR binds PopZ only in its phosphorylated state.

I recommend reanalyzing the data using the partition coefficient method, with at least 50 droplets per condition, including two additional biological replicates. Additionally, depositing the droplet images (and all other images) to a platform like Zenodo or Dropbox would enhance transparency. An orthogonal approach to demonstrate binding specificity upon phosphorylation could be explored using NMR.

Response: We performed additional repetitions of this experiment in the newly revised manuscript. We now show quantify n=50 droplets in each of three separate replicates. Also, we have replaced the images in Figure 3b with new images, which show more droplets, more detailed ring fluorescence, and the surrounding pool of unbound CpdR-GFP. Raw images will be deposited as "Source Data" according to editor's instruction.

Minor comment: y-axis labeling is missing for Fig. 3d, and Fig. 3b is not referred to in the main text.

Response: The Y-axis in Fig. 3d is now labeled: "fold difference", and indicates the ratios of the CpdR-GFP and CpdR_{D51A}-GFP fluorescence intensities within condensates to outside of condensates. We refer to Fig 3b in the main text in the following sentences of the results section: 'We asked if CpdR can also interact with macromolecular scaffolds comprised of full-length PopZ, and if this interaction can be influenced by CpdR phosphorylation. To do this, we generated phase-separated condensates of purified full-length PopZ¹⁰ and demonstrated that CpdR-GFP but not GFP alone partitions into the condensates (Fig. 3b).'